# Transition role of entangled data in quantum machine learning

Xinbiao Wang[1,2,3], Yuxuan Du [3,4] ✉, Zhuozhuo Tu[5], Yong Luo [1,2] ✉,
Xiao Yuan [6,7] & Dacheng Tao [4] ✉

Entanglement serves as the resource to empower quantum computing. Recent progress has highlighted its positive impact on learning quantum dynamics, wherein the integration of entanglement into quantum operations or measurements of quantum machine learning (QML) models leads to substantial reductions in training data size, surpassing a specified prediction error threshold. However, an analytical understanding of how the entanglement degree in data affects model performance remains elusive. In this study, we address this knowledge gap by establishing a quantum no-free-lunch (NFL) theorem for learning quantum dynamics using entangled data. Contrary to previous findings, we prove that the impact of entangled data on prediction error exhibits a dual effect, depending on the number of permitted measurements. With a sufficient number of measurements, increasing the entanglement of training data consistently reduces the prediction error or decreases the required size of the training data to achieve the same prediction error. Conversely, when few measurements are allowed, employing highly entangled data could lead to an increased prediction error. The achieved results provide critical guidance for designing advanced QML protocols, especially for those tailored for execution on early-stage quantum computers with limited access to quantum resources.

Quantum entanglement, an extraordinary characteristic of the quantum realm, drives the superiority of quantum computers beyond classical computers[1]. Over the past decade, diverse quantum algorithms leveraging entanglement have been designed to advance cryptography[2,3] and optimization[4–8], delivering runtime speedups over classical approaches. Motivated by the exceptional abilities of quantum computers and the astonishing success in machine learning, a nascent frontier known as quantum machine learning (QML) has emerged[9–15], seeking to outperform classical models in specific learning tasks[16–25]. Substantial progress has been made in this field, exemplified by the introduction of QML

protocols that offer provable advantages in terms of query or sample complexity for learning quantum dynamics[26–31], as a fundamental problem toward understanding the laws of nature. Most of these protocols share a common strategy to gain advantages: the incorporation of entanglement into quantum operations and measurements, leading to reduced complexity. Nevertheless, an overlooked aspect in prior works is the impact of incorporating entanglement in quantum input states, or entangled data, on the advancement of QML in learning quantum dynamics. Due to the paramount role of data in learning[32–37] as well as entanglement in quantum computing, addressing this question will significantly

[1]Institute of Artificial Intelligence, School of Computer Science, Wuhan University, Hubei 430072, China. [2]National Engineering Research Center for Multimedia Software, Wuhan University, Hubei 430072, China. [3]JD Explore Academy, Beijing 101111, China. [4]School of Computer Science and Engineering, Nanyang Technological University, Singapore 639798, Singapore. [5]School of Computer Science, Faculty of Engineering, University of Sydney, Sydney, NSW 2008, Australia. [6]Center on Frontiers of Computing Studies, Peking University, Beijing 100871, China. [7]School of Computer Science, Peking University, Beijing 100871, China. ✉e-mail: duyuxuan123@gmail.com; yluo180@gmail.com; dacheng.tao@ntu.edu.sg

enhance our comprehension of the capabilities and limitations of QML models.

A fundamental concept in machine learning that characterizes the capabilities of learning models in relation to datasets is the no-free-lunch (NFL) theorem[38–41]. The NFL theorem yields a key insight: regardless of the optimization strategy employed, the ultimate performance of models is contingent upon the size and types of training data. This observation has spurred recent breakthroughs in large language models, as extensive and meticulously curated training data consistently yield superior results[42–46]. In this regard, establishing the quantum NFL theorem enables us to elucidate the specific impact of entangled data on the efficacy of QML models in learning quantum dynamics. Concretely, the achieved theorem can shed light on whether the utilization of entangled data empowers QML models to achieve comparable or even superior performance compared to low-entangled or unentangled data, while simultaneously reducing the sample complexity required. Although initial attempts[47,48,49] have been made to establish quantum NFL theorems, they have relied on infinite query complexity, thus failing to address our concerns adequately (see Supplementary Note 1 and Supplementary Note 2 for details). Building upon prior findings on the role of entanglement and the classical NFL theorem, a reasonable speculation is that high-entangled data contributes to the improved performance of QML models associated with the reduced sample complexity, albeit at the cost of using extensive quantum resources to prepare such data that may be unaffordable in the early stages of quantum computing[50].

In this study, we *negate* the above speculation and exhibit the *transition role* of entangled data when QML models incoherently learn quantum dynamics, as shown in Fig. 1. In the incoherent learning scenario, the quantum learner is restricted to utilizing datasets with varying degrees of entanglement to operate on an unknown unitary and inferring its dynamics using the finite measurement outcomes collected under the projective measurement, differing from ref. 48 in learning problems and training data. The entangled data refers to quantum states that are entangled with a reference system, with the degree of entanglement quantitatively characterized by the Schmidt rank $r$. We rigorously show that within the context of NFL, the entangled data has a *dual effect* on the prediction error according to the number of measurements $m$ allowed. Particularly, with sufficiently

large $m$, increasing $r$ can consistently reduce the required size of training data for achieving the same prediction error. On the other hand, when $m$ is small, the train data with large $r$ not only requires a significant volume of quantum resources for states preparation, but also amplifies the prediction error. As a byproduct, we prove that the lower bound of the query complexity for achieving a sufficiently small prediction error matches the optimal lower bound for quantum state tomography with nonadaptive measurements. To cover a more generic learning setting, we consider the problem of dynamic learning under arbitrary observable by using $\ell$-outcome positive-operator valued measure (POVM) to collect the measurement output. This setting covers the shadow-based learning models[26,51,52]. We show that the transition role still holds for arbitrary POVM and increasing the possible outcomes $\ell$ could significantly reduce the query complexity. Numerical simulations are conducted to support our theoretical findings. In contrast to the previous understanding that entanglement mostly confers benefits to QML in terms of sample complexity, the transition role of entanglement identified in this work deepens our comprehension of the relation between quantum information and QML, which facilitates the design of QML models with provable advantages.

## Results

We first recap the task of learning quantum dynamics. Let $\boldsymbol{U} \in \mathbb{SU}(2^n)$ be the target unitary and $\boldsymbol{O} \in \mathbb{C}^{2^n \times 2^n}$ be the observable which is a Hermitian matrix acting on an $n$-qubit quantum system. Here we specify the observable as the projective measurement $\boldsymbol{O} = |\boldsymbol{o}\rangle\langle\boldsymbol{o}|$ since any observable reads out the classical information from the quantum system via their eigenvectors. The goal of the quantum dynamics learning is to predict the functions of the form

$$f_{\boldsymbol{U}}(\boldsymbol{\psi}) = \text{Tr}(\boldsymbol{O}\boldsymbol{U}|\boldsymbol{\psi}\rangle\langle\boldsymbol{\psi}|\boldsymbol{U}^{\dagger}), \tag{1}$$

where $|\boldsymbol{\psi}\rangle$ is an $n$-qubit quantum state living in a $2^n$-dimensional Hilbert space $\mathcal{H}_{\mathcal{X}}$. This task can be done by employing the training data $\mathcal{S}$ to construct a unitary $\boldsymbol{V}_{\mathcal{S}}$, i.e., the learned hypothesis has the form of $h_{\mathcal{S}}(\boldsymbol{\psi}) = \text{Tr}(\boldsymbol{O}\boldsymbol{V}_{\mathcal{S}}|\boldsymbol{\psi}\rangle\langle\boldsymbol{\psi}|\boldsymbol{V}_{\mathcal{S}}^{\dagger})$, which is expected to accurately approximate $f_{\boldsymbol{U}}(\boldsymbol{\psi})$ for the unseen data. While the learned unitary acts on an $n$-qubit system $\mathcal{H}_{\mathcal{X}}$, the input state could be entangled with a reference

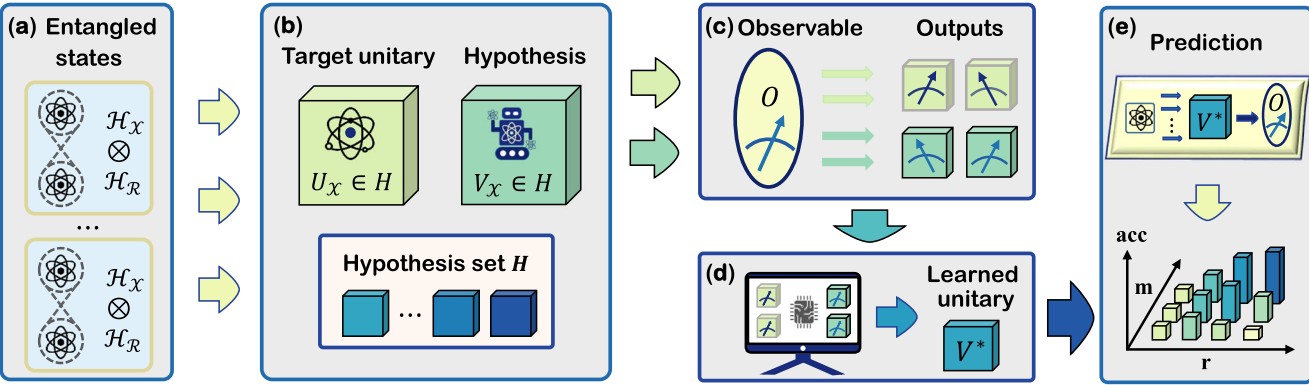

**Fig. 1 | Illustration of quantum NFL setting with the entangled data.** The goal of the quantum learner is to learn a unitary $\boldsymbol{V}_{\mathcal{X}}$ that can accurately predict the output of the target unitary $\boldsymbol{U}_{\mathcal{X}}$ under a fixed observable $\boldsymbol{O}$, where the subscript $\mathcal{X}$ refers to the quantum system in which the operator $\boldsymbol{O}$ act on. The learning process is as follows. **a** A total number of $N$ entangled bipartite quantum states living in Hilbert space $\mathcal{H}_{\mathcal{X}} \otimes \mathcal{H}_{\mathcal{R}}$ ($\mathcal{R}$ denotes the reference system) are taken as inputs, dubbed *entangled data*. **b** Quantum learner proceeds incoherent learning. The entangled data *separately* interacts with the target unitary $\boldsymbol{U}_{\mathcal{X}}$ (agnostic) and the candidate hypothesis $\boldsymbol{V}_{\mathcal{X}}$ extracted from the same Hypothesis set $H$. **c** The quantum learner is restricted to leverage the finite measured outcomes of the observable $\boldsymbol{O}$ on the output states of $\boldsymbol{U}_{\mathcal{X}}$ and $\boldsymbol{V}_{\mathcal{X}}$ to conduct learning. **d** A classical computer is exploited to infer $\boldsymbol{V}$ that best estimates $\boldsymbol{U}_{\mathcal{X}}$ according to the measurement outcomes. For example, in the case of variational quantum algorithms, the classical computer serves as an optimizer to update the tunable parameters of the ansatz $\boldsymbol{V}_{\mathcal{X}}$. **e** The learned unitary $\boldsymbol{V}$ is used to predict the output of unseen quantum states in Hilbert space $\mathcal{H}_{\mathcal{X}}$ under the evolution of the target unitary $\boldsymbol{U}_{\mathcal{X}}$ and the measurement of $\boldsymbol{O}$. A large Schmidt rank $r$ can enhance the prediction accuracy when combined with a large number of measurements $m$, but may lead to a decrease in accuracy when $m$ is small.

system $\mathcal{H}_{\mathcal{R}}$, i.e., $|\boldsymbol{\psi}\rangle \in \mathcal{H}_{\mathcal{X}} \otimes \mathcal{H}_{\mathcal{R}}$. We suppose that all input states have the same Schmidt rank $r \in \{1, \cdots, 2^n\}$. Then the response of the state $|\boldsymbol{\psi}_j\rangle$ is given by the measurement output $\boldsymbol{o}_j = \sum_{k=1}^m \boldsymbol{o}_{jk}/m$, where $m$ is the number of measurements and $\boldsymbol{o}_{jk}$ is the output of the $k$-th measurement of the observable $\boldsymbol{O}$ on the output quantum state $(\boldsymbol{U} \otimes \mathbb{I}_{\mathcal{R}})|\boldsymbol{\psi}_j\rangle$. In this manner, the training data with $N$ examples takes the form $\mathcal{S} = \{(|\boldsymbol{\psi}_j\rangle, \boldsymbol{o}_j) : |\boldsymbol{\psi}_j\rangle \in \mathcal{H}_{\mathcal{X}} \otimes \mathcal{H}_{\mathcal{R}}, \mathbb{E}[\boldsymbol{o}_j] = u_j\}_{j=1}^N$ with $u_j = \mathrm{Tr}((\boldsymbol{U}^\dagger \boldsymbol{O} \boldsymbol{U} \otimes \mathbb{I}_{\mathcal{R}})|\boldsymbol{\psi}_j\rangle\langle\boldsymbol{\psi}_j|)$ being the expectation value of the observable $\boldsymbol{O}$ on the state $(\boldsymbol{U} \otimes \mathbb{I}_{\mathcal{R}})|\boldsymbol{\psi}_j\rangle$ and $N$ being the size of the training data. Notably, in quantum dynamics learning, sample complexity refers to the size of training data $N$, or equivalently, the number of quantum states in the training data; query complexity refers to the total number of queries of the explored quantum system, i.e., the production of sample complexity and the number of measurements $Nm$.

The risk function is a crucial measure in statistical learning theory to quantify how well the hypothesis function $\mathrm{h}_{\mathcal{S}}$ performs in predicting $f_{\boldsymbol{U}}$, defined as

$$R_{\boldsymbol{U}}(\boldsymbol{V}_{\mathcal{S}}) = \int d\boldsymbol{\psi}\big(f_{\boldsymbol{U}}(\boldsymbol{\psi}) - \mathrm{h}_{\mathcal{S}}(\boldsymbol{\psi})\big)^2, \tag{2}$$

where the integral is over the uniform Haar measure $d\boldsymbol{\psi}$ on the state space. Intuitively, $R_{\boldsymbol{U}}(\boldsymbol{V}_{\mathcal{S}})$ amounts to the average square error distance between the true output $f(\boldsymbol{\psi})$ and the hypothesis output $\mathrm{h}_{\mathcal{S}}(\boldsymbol{\psi})$. Moreover, we follow the treatments in ref. [48] choosing the Haar unitary as the target unitary. Additionally, we construct a sampling rule of the training input states which approximates the uniform distribution of all entangled states with Schmidt rank $r$ (refer to Supplementary Note 2).

Under the above setting, we prove the following quantum NFL theorem in learning quantum dynamics, where the formal statement and proof are deferred to Supplementary Note 3.

**Theorem 1.** (Quantum NFL theorem in learning quantum dynamics, informal). Following the settings in Eq. (1), suppose that the training error of the learned hypothesis on the training data $\mathcal{S}$ is less than $\varepsilon = \mathcal{O}(1/2^n)$. Then the lower bound of the averaged prediction error in Eq. (2) yields

$$\mathbb{E}_{\boldsymbol{U}, \mathcal{S}} R_{\boldsymbol{U}}(\boldsymbol{V}_{\mathcal{S}}) \geq \Omega\left(\frac{\bar{\varepsilon}^2}{4^n}\left(1 - \frac{N \cdot \min\{m/(2^n r c_1), rn\}}{2^n c_2}\right)\right),$$

where $c_1 = 128/\bar{\varepsilon}^2$, $c_2 = \min\{(1 - 2\bar{\varepsilon})^2, (64\bar{\varepsilon}^2 - 1)^2\}$, $\bar{\varepsilon} = \Theta(2^n \varepsilon)$, and the expectation is taken over all target unitary $\boldsymbol{U}$, entangled states $|\boldsymbol{\psi}_j\rangle$ and measurement outputs $\boldsymbol{o}_j$.

The achieved results indicate the transition role of the entangled data in determining the prediction error. Particularly, when a sufficient number of measurements $m$ is allowed such that the Schmidt rank $r$ obeys $r < \sqrt{m/(c_1 2^n n)}$, the minimum term in the achieved lower bound refers to $Nrn$ and hence increasing $r$ can constantly decrease the prediction error. Accordingly, in the two extreme cases of $r = 1$ and $r = 2^n$, achieving zero averaged risk requires $N = 2^n c_2/n$ and $N = 1$ training input states, where the latter achieves an exponential reduction in the number of training data compared with the former. This observation implies that the entangled data empower QML with provable quantum advantage, which accords with the achieved results of ref. [48] in the ideal coherent learning protocol with infinite measurements.

By contrast, in the scenario with $r \geq \sqrt{m/(c_1 2^n n)}$, increasing $r$ could enlarge the prediction error. This result indicates that the entangled data can be harmful to achieving quantum advantages, which *contrasts with previous results* where the entanglement (e.g., entangled operations or measurements) is believed to contribute to the quantum advantage[48,53–55]. This counterintuitive phenomenon stems from the fact that when incoherently learning quantum

dynamics, information obtained from each measurement decreases with the increased $r$ and hence a small $m$ is incapable of extracting all information of the target unitary carried by the entangled state.

Another implication of Theorem 1 is that although the number of measurements $m$ contributes to a small prediction error, it is *not decisive to the ultimate performance* of the prediction error. Specifically, when $m \geq r^2 c_1 2^n n$, further increasing $m$ could not help decrease the prediction error which is determined by the entanglement and the size of the training data, i.e., $r$ and $N$. Meanwhile, at least $r^2 c_1 2^n n$ measurements are required to fully utilize the power of entangled data. These results suggest that the value of $m$ should be adaptive to $r$ to pursue a low prediction error.

We next comprehend the scenario in which the lower bound of averaged risk in Theorem 1 reaches zero and correlate with the results in quantum state learning and quantum dynamics learning[26,27,29,30,56,57]. In particular, the main focus of those studies is proving the minimum query complexity of the target unitary to warrant zero risk. The results in Theorem 1 indicate that the minimum query complexity is $Nm = \Omega(4^n r c_1 c_2)$, implying the proportional relation between the entanglement degree $r$ and the query complexity. Notably, *this lower bound is tighter than that achieved in* ref. [26] in the same setting. The achieved results in terms of query complexity are also non-trivial, as previous works show that query complexity can benefit from using entanglement in quantum data[58,59] and quantum measurements[26,30]. The advance of our results stems from the fact that ref. [26] simply employs Holevo's theorem to give an upper bound on the extracted information in a single measurement, while our bound integrates more refined analysis such as the consideration of Schmidt rank $r$, the direct use of a connection between the mutual information of the target unitary $\boldsymbol{U}$ and the measurement outputs $\boldsymbol{o}_j$, and the KL-divergence of related distributions (refer to Supplementary Note 3 for more details). Moreover, the adopted projective measurement $\boldsymbol{O}$ in Eqn. (1) hints that the learning task explored in our study amounts to learning a pure state $\boldsymbol{U}^\dagger \boldsymbol{O} \boldsymbol{U}$. From the perspective of state learning, *the derived lower bound in Theorem 1 is optimal for the nonadaptive measurement with a constant number of outcomes*[60]. Taken together, while the entangled data hold the promise of gaining advantages in terms of the sample complexity for achieving the same level of prediction error, they may be inferior to the training data without entanglement in terms of query complexity.

The transition role of entanglement explained above leads to the following construction rule of quantum learning models. First, when a large number of measurements is allowed, the entangled data is encouraged to be used for improving the prediction performance. To this end, initial research efforts[61–66], which develop effective methods for preparing and storing entangled states, may contribute to QML. Second, when the total number of measurements is limited, it is advised to refrain from using entangled data for learning quantum dynamics.

*Remark.* (i) The training error scaling $\varepsilon = \mathcal{O}(1/2^n)$ in Theorem 1 and the factor of the achieved lower bound $\bar{\varepsilon}^2/4^n$ comes from the consideration of average performance over Haar unitaries where the expectation value of observable $\boldsymbol{O}$ scales as $\mathrm{Tr}(\boldsymbol{O})/2^n$ (Refer to Supplementary Note 2). (ii) The results of the transition role for entangled data achieved in Theorem 1 can be generalized to the mixed states because the mixed state can be produced by taking the partial trace of a pure entangled state.

In a more generic learning setting, the observable used in the target function defined in Eqn. (1) and the measurement used for collecting the response of training data $\boldsymbol{o}$ could be arbitrary and varied. In particular, we consider that the observable $\boldsymbol{O}$ defined in Eqn. (1) could be arbitrary Hermitian operator satisfying $\|\boldsymbol{O}\|_1 \leq \infty$. The response $\boldsymbol{a}_j$ for given input states $|\boldsymbol{\psi}_j\rangle$ could be obtained from measuring the output states on system $\mathcal{X}$ with $\ell$-outcome POVM. The training dataset in this case refers to $\mathcal{S}_\ell = \{(|\boldsymbol{\psi}_j\rangle, \boldsymbol{a}_j) : |\boldsymbol{\psi}_j\rangle \in \mathcal{H}_{\mathcal{X}\mathcal{R}}, \boldsymbol{a}_j = (\boldsymbol{a}_{j1}, \cdots, \boldsymbol{a}_{jm}), \boldsymbol{a}_{jk} \in \{z_1, \cdots, z_\ell\}\}_{j=1}^N$,

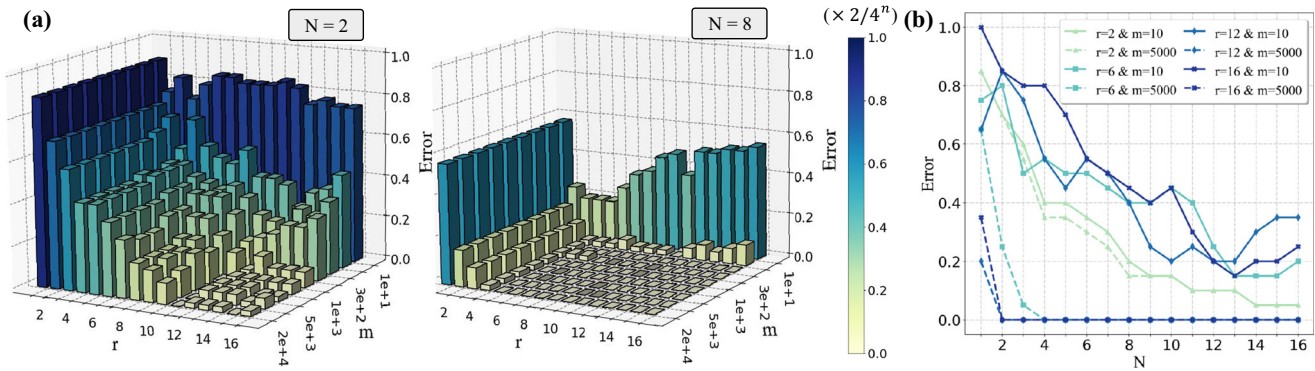

**Fig. 2 | Simulation results of quantum NFL theorem when incoherently learning quantum dynamics. a** The averaged prediction error with a varied number of measurements $m$ and Schmidt rank $r$ when $N = 2$ and $N = 8$. The $z$-axis refers to the averaged prediction error defined in Eq (1). **b** The averaged prediction error with the varied sizes of training data. The label "$r = a \& m = b$" refers that the Schmidt rank is $a$ and the number of measurements is $b$. The label "($\times 2/4^n$)" refers that the plotted prediction error is normalized by a multiplier factor $2/4^n$.

where $|\psi_j\rangle$ refers to the entangled states with Schmidt rank $r$, $a_j$ is the $m$-measurement outputs with $\ell$-outcome POVM, and $\{z_i\}_{i=1}^{\ell}$ is the $\ell$ possible outcomes of the employed POVM. In this case, denoting the learned unitary as $V_{S_\ell}$, we get the following quantum NFL theorem in learning quantum dynamics for generic measurements, where the formal statement and proof are deferred to Supplementary Note 4.

**Theorem 2.** (Quantum NFL theorem in learning quantum dynamics for generic measurements, informal) Following the settings in Eq. (1) with arbitrary $O$ satisfying $\|O\|_1 \leq \infty$, suppose the learned hypothesis is learned from training data $S_\ell$. Then the lower bound of the averaged prediction error in Eqn. (2) yields

$$\mathbb{E}_{U,S_\ell} R_U(V_{S_\ell}) \geq \varepsilon^2 \left(1 - \frac{N \cdot \min\{4m/r, 6m\ell/2^n r, rn\}}{\log(|\mathcal{X}_{2\varepsilon}(O)|)}\right)$$

where $|\mathcal{X}_{2\varepsilon}(O)|$ refers to the model complexity and only depends on $\varepsilon$ and the employed observable $O$. For projective measurement $O = |o\rangle\langle o|$, $\log(|\mathcal{X}_{2\varepsilon}(O)|) = 2^n c_2$ is given in the denominator of the achieve lower bound in Theorem 1.

The achieved results in Theorem 2 deliver three implications. First, the transition role of entangled data still holds for arbitrary observable and POVM. In particular, no matter how large the number of possible outcomes of POVM $\ell$ is, increasing the Schmidt rank will decrease the prediction error as long as the number of measurements $m$ satisfies $\min\{4m/r, 6m\ell/2^n r\} \leq rn$, and increase the prediction error otherwise. Second, when the observable is projective measurement and the number of possible outcomes $\ell$ is of constant order, the achieved result in Theorem 2 reduces to the results achieved in Theorem 1 for the case of employing projective measurement up to a constant factor. Third, increasing the number of possible outcomes of POVM $\ell$ can exponentially reduce the number of measurements required to achieve the same level of prediction error. Particularly, considering two extreme cases of the possible outcomes of POVM $\ell$ being constant scaling $\Theta(1)$ and exponential scaling $\Theta(2^n)$, achieving the same level of prediction error requires the query complexity scaling with the order of $2^n r \log(|\mathcal{X}_{2\varepsilon}(O)|)$ and $r \log(|\mathcal{X}_{2\varepsilon}(O)|)$, where the latter case achieves an exponential reduction in terms of the query complexity.

### Numerical results
We conduct numerical simulations to exhibit the transition role of entangled data, the effect of the number of measurements, and the training data size in determining the prediction error. The omitted construction details and results are deferred to Supplementary Note 5.

We focus on the task of learning an $n$-qubit unitary under a fixed projective measurement $O = (|O\rangle\langle O|)^{\otimes n}$. The number of qubits is $n = 4$. The target unitary $U_X$ is chosen uniformly from a discrete set $\{U_i\}_{i=1}^M$, where $M = 2^n$ refers to the set size and the operators $U_i^\dagger O U_j$ with $U_j$ in this set are orthogonal such that the operators $U_j^\dagger O U_j$ are well distinguished. The entangled states in $S$ is uniformly sampled from the set $\{\sum_{j=1}^r \sqrt{c_j} U_j |O\rangle \otimes |\xi_j\rangle \mid (\sqrt{c_1}, \cdots, \sqrt{c_r})^\top \in \mathbb{SU}(r), |\xi_j\rangle \in \mathbb{SU}(2^n)\}$. The size of training data is $N \in \{1, 2, \cdots, 16\}$ and the Schmidt rank takes $r = \{2^0, \cdots, 2^4\}$. The number of measurements takes $m \in \{10, 100, 300, \cdots, 5000, 20000\}$. We record the averaged prediction error by learning four different 4-qubit unitaries for 10 training data.

The simulation results are displayed in Fig. 2. Particularly, Fig. 2a shows that for both the cases of $N = 2$ and $N = 8$, the prediction error constantly decreases with respect to an increased number of measurements $m$ and increased Schmidt rank $r$ when the number of measurements is large enough, namely $m > 1000$. On the other hand, for a small number of measurements with $m \leq 100$ in the case of $N = 8$, as the Schmidt rank is continually increased, the averaged prediction error initially decreases and then increases after the Schmidt rank surpasses a critical point which is $r = 3$ for $m = 10$ and $r = 4$ for $m = 100$. This phenomenon accords with the theoretical results in Theorem 1 in the sense that the entangled data play a transition role in determining the prediction error for a limited number of measurements. This observation is also verified in Fig. 2b for the varied sizes of training data, where for the small measurement times $m = 10$, increasing the Schmidt rank could be not helpful for decreasing the prediction error. By contrast, a large training data size consistently contributes to a small prediction error, which echoes with Theorem 1.

## Discussion
In this study, we exploited the effect of the Schmidt rank of entangled data on the performance of learning quantum dynamics with a fixed observable. Within the framework of the quantum NFL theorem, our theoretical findings reveal the transition role of entanglement in determining the ultimate model performance. Specifically, increasing the Schmidt rank below a threshold controlled by the number of measurements can enhance model performance, whereas surpassing this threshold can lead to a deterioration in model performance. Our analysis suggests that a large number of measurements is the precondition to use entangled data to gain potential quantum advantages. In addition, our results demystify the negative role of entangled data in the measure of query complexity. Last, as with the classical NFL theorem, we prove that increasing the size of the training data always contributes to a better performance in QML.

Our results motivate several important issues and questions needed to be further investigated. The first research direction is exploring whether the transition role of entangled data exists for other QML tasks such as learning quantum unitaries or learning quantum channels with the response being measurement output[26,30,55,67–76]. These questions can be considered in both the coherent and incoherent learning protocols, which are determined by whether the target and model system can coherently interact and whether quantum information can be shared between them. Obtaining such results would have important implications for using QML models to solve practical tasks with provable advantages.

A another research direction is inquiring whether there exists a similar transition role when exploiting entanglement in quantum dynamics and measurements through the use of an ancillary quantum system. The answer for the case of entangled measurement has been given under many specific learning tasks[26,29,30,77] where the learning protocols with entangled measurements are shown to achieve an exponential advantage over those without in terms the access times to the target unitary. This quantum advantage arises from the powerful information-extraction capabilities of entangled measurements. In this regard, it is intriguing to investigate the effect of quantum entanglement on model performance when entanglement is introduced in both the training states and measurements, as entangled measurements offer a potential solution to the negative impact of entangled data resulting from insufficient information extraction via weak projective measurements. A positive result could further enhance the quantum advantage gained through entanglement exploitation.

## Methods

Here we first outline the proof strategy that establishes the lower bound of the averaged prediction error in Theorem 1. Then we present an intuitive explanation of the transition role of entangled data according to the achieved numerical results.

### Proof sketch

The backbone of the proof refers to Fano's method, which is widely used to derive the lower bound of prediction error in classical learning theory[78]. This method involves the following three parts. Part (I): The space of the target dynamics $\mathcal{U} = \{\boldsymbol{U} \in \mathbb{SU}(d)\}$ is discretized into a $2\varepsilon$-packing $\mathcal{M}_{2\varepsilon} = \{\boldsymbol{U}_{x'}\}_{x'=1}^{|\mathcal{M}_{2\varepsilon}|}$ such that the dynamics within $\mathcal{M}_{2\varepsilon}$ are sufficiently distinguishable under a distance metric related to the target function in Eq. (1). Part (II): The dynamics learning problem in Eq. (1) is translated to the hypothesis testing problem related to the $2\varepsilon$-packing $\mathcal{M}_{2\varepsilon}$. Such a hypothesis testing problem amounts to a communication protocol between two parties, namely Alice and Bob. Particularly, Alice chooses an element $X$ of $\{1, \cdots, |\mathcal{M}_{2\varepsilon}|\}$ uniformly at random and employs the corresponding unitary $\boldsymbol{U}_X$ to construct the training data $\mathcal{S} = \{|\boldsymbol{\psi}_j\rangle, \boldsymbol{o}_j^{(X)}\}_{j=1}^N$ with $|\boldsymbol{\psi}_j\rangle$ being the randomly sampled entangled input state and $\boldsymbol{o}_j$ being the associated measurement output of the state $(\boldsymbol{U}_X \otimes \mathbb{I})|\boldsymbol{\psi}_j\rangle$ under the projective measurement $\boldsymbol{O}$. Bob's goal is to retrieve the information of $X$ from the discrete set $\{1, \cdots, |\mathcal{M}_{2\varepsilon}|\}$ based on $\mathcal{S}$. The inferred index by Bob is denoted by $\hat{X}$. This leads to the hypothesis testing problem with the null hypothesis $\hat{X} \neq X$. In this regard, we demonstrate that the averaged prediction error is greater than a quantity related to the error probability $\mathbb{P}(\hat{X} \neq X)$ of the hypothesis testing problem, namely $\mathbb{E}_{\boldsymbol{U}, \mathcal{S}} \mathsf{R}_{\boldsymbol{U}}(\boldsymbol{V}_{\mathcal{S}}) \geq \mathbb{E}_{X, \mathcal{S}} \varepsilon^2 \mathbb{P}(\hat{X} \neq X)$. This provides the theoretical guarantee of reducing the learning problem to the hypothesis testing problem. Part (III): Fano's inequality is utilized to establish an upper bound on the error probability $\mathbb{P}(\hat{X} \neq X)$ of the hypothesis testing problem, i.e.,

$$\mathbb{P}(\hat{X} \neq X) \geq 1 - \frac{I(X; \hat{X}) + \log 2}{\log(|\mathcal{M}_{2\varepsilon}|)}, \quad (3)$$

which is dependent on two factors: the cardinality of the $2\varepsilon$-packing $\mathcal{M}_{2\varepsilon}$, and the mutual information $I(X; \hat{X})$ between the target index $X$ and the estimated index $\hat{X}$.

To summarize, Fano's method reduces the challenging problem of lower bounding the prediction error to separately lower bounding the packing cardinality and upper bounding the mutual information, which we could develop techniques for tackling. In particular, we obtain the lower bound of the $2\varepsilon$-packing cardinality $|\mathcal{M}_{2\varepsilon}|$ by employing the probabilistic argument to show the existence of a large but well-separated collection of quantum dynamics (i.e., the $2\varepsilon$-packing $\mathcal{M}_{2\varepsilon}$) under a metric dependent on the observable $\boldsymbol{O}$.

We establish an upper bound for the mutual information $I(X; \hat{X})$ by considering two cases: one with a small number of measurements and another with a sufficiently large number of measurements. For the former case, the mutual information $I(X; \hat{X})$ is upper bounded by a quantity involving the KL-divergence between the probability distributions of the measurement output $\boldsymbol{o}_j^{(x)}$ related to various index $x$, which has the order of $\mathcal{O}(Nmd\varepsilon^2/r)$ in the average case. On the other hand, while the mutual information $I(X; \hat{X})$ cannot grow infinitely with the number of measurements, we derive another upper bound of $I(X; \hat{X})$ with the mutual information $I(X; \{(\boldsymbol{U}_X \otimes \mathbb{I})|\boldsymbol{\psi}_j\rangle\}_{j=1}^N)$ for the case of a sufficiently large number of measurements which could extract the maximal amount of information from each output state. In this regard, the mutual information is upper bounded by an $m$-independent quantity with the order of $\mathcal{O}(rn)$. This leads to the final upper bound of the mutual information $\min\{\mathcal{O}(Nmd\varepsilon^2/r), \mathcal{O}(rn)\}$ where the Schmidt rank $r$ plays the opposite role in the two scenarios with the various number of measurements allowed, resulting in the transition role of entangled data.

Taken all together, we can obtain the lower bound of the averaged prediction error in Theorem 1.

**Remark.** While many similar pipelines involving the utilization of Fano's inequality have been used for obtaining the lower bound of sample complexity in quantum state learning tasks, we give a detailed explanation about how our results differ from previous studies in Supplementary Note 1E.

### Intuitive explanations based on numerical results

We now give an intuitive explanation about the transition role of the entangled data based on the numerical simulations.

Before elucidating, we first detail the construction rule of the target unitaries set and the entangled input states. Particularly, to construct the set consisting of well-distinguished target unitaries under the distance metric related to the observable $\boldsymbol{O} = (|\boldsymbol{0}\rangle\langle\boldsymbol{0}|)^{\otimes n}$, one way is to choose the unitaries $\boldsymbol{U}_j$ in $\mathbb{SU}(2^n)$ at uniformly random such that the target operators $\boldsymbol{U}_j^\dagger \boldsymbol{O} \boldsymbol{U}_j = |\boldsymbol{e}_j\rangle\langle\boldsymbol{e}_j|$ are mutually orthogonal. In this regard, learning the target unitary under the observable $\boldsymbol{O}$ is equivalent to identifying the unknown index $k^*$ corresponding to the target operator $\boldsymbol{U}_{k^*}^\dagger \boldsymbol{O} \boldsymbol{U}_{k^*} = |\boldsymbol{e}_{k^*}\rangle\langle\boldsymbol{e}_{k^*}|$. The entangled input states have the form of $|\boldsymbol{\psi}_j\rangle = \sum_{k=1}^r \sqrt{c_{jk}} |\boldsymbol{\xi}_{jk}\rangle_{\mathcal{X}} |\boldsymbol{\zeta}_{jk}\rangle_{\mathcal{R}}$ where the Schmidt coefficients $\{c_{jk}\}$ satisfy $\sum_{k=1}^r c_{jk} = 1$. As the target unitary $\boldsymbol{U}_{k^*}$ acts on the quantum system $\mathcal{X}$, the identification of the corresponding index $k^*$ solely depends on the partial trace of the entangled states, i.e., $\sigma_j := \mathrm{Tr}_{\mathcal{R}}(|\boldsymbol{\psi}_j\rangle\langle\boldsymbol{\psi}_j|) = \sum_{k=1}^r c_{jk} |\boldsymbol{\xi}_{jk}\rangle\langle\boldsymbol{\xi}_{jk}|_{\mathcal{X}}$. To this end, we consider that the states $|\boldsymbol{\xi}_{jk}\rangle$ in the reduced states are sampled from the computational basis $\{|\boldsymbol{e}_i\rangle\}_{i=1}^{2^n}$ and the coefficient vector $\boldsymbol{c}_j = (\sqrt{c_{j1}}, \cdots, \sqrt{c_{jr}})$ is sampled from the Haar distribution in the $r$-dimensional Hilbert space $\mathcal{H}_r$. In this manner, we construct $\mathcal{S} = \{|\boldsymbol{\psi}_j\rangle, \boldsymbol{o}_j\}_{j=1}^N$ and use it to learn the unknown index $k^*$ by solving the following minimization

problem

$$\hat{k} = \arg\min_{k \in [2^n]} \sum_{j=1}^{N} \left( \boldsymbol{o}_j^{(k)} - \boldsymbol{o}_j \right)^2, \tag{4}$$

where $[2^n]$ refers to the set $\{1, \cdots, 2^n\}$ and $\boldsymbol{o}_j^{(k)}$ is the collected measurement outputs by applying the observable $\boldsymbol{U}_k^\dagger \boldsymbol{O} \boldsymbol{U}_k$ with $\boldsymbol{U}_k \in \{\boldsymbol{U}_{k'}\}_{k' \in [2^n]}$ to input states $|\boldsymbol{\psi}_j\rangle$.

The successful identification of the target index $k^*$ relies on the satisfaction of two key conditions:

1. The states set $\{|\boldsymbol{\xi}_{ji}\rangle\langle\boldsymbol{\xi}_{ji}|\}_{j,i=1}^{N,r}$ contains the target operator $\boldsymbol{U}_{k^*}^\dagger \boldsymbol{O} \boldsymbol{U}_{k^*} = |\boldsymbol{e}_{k^*}\rangle\langle\boldsymbol{e}_{k^*}|$.
2. The measurement outputs $\{\boldsymbol{o}_j\}_{j=1}^N$ closely approximate the corresponding Schmidt coefficient $c_{k^*}$ of the operator $\boldsymbol{U}_{k^*}^\dagger \boldsymbol{O} \boldsymbol{U}_{k^*} = |\boldsymbol{e}_{k^*}\rangle\langle\boldsymbol{e}_{k^*}| \in \{|\boldsymbol{\xi}_{ji}\rangle\langle\boldsymbol{\xi}_{ji}|\}_{j,i=1}^{N,r}$.

The first condition ensures that the measurement outputs $\{\boldsymbol{o}_j\}_{j=1}^N$ are non-zero, enabling the identification of the target index $k^*$. Moreover, if the target operator $\boldsymbol{U}_{k^*}^\dagger \boldsymbol{O} \boldsymbol{U}_{k^*}$ is not included in $\{|\boldsymbol{\xi}_{ji}\rangle\langle\boldsymbol{\xi}_{ji}|\}_{j,i=1}^{N,r}$, measuring the output states with observable $\boldsymbol{O}$ will always yield zero for all $k \in [2^n]$ such that any $k \in [2^n]$ is the solution of Eqn. (4).

The second condition ensures that the non-zero measurement outcomes can closely approximate the ground truth to facilitate the identification of the target index $k^*$. In this regard, the states set $\{|\boldsymbol{\xi}_{ji}\rangle\langle\boldsymbol{\xi}_{ji}|\}_{j,i=1}^{N,r}$, which associates with highly entangled input states (a large Schmidt rank $r$), is more likely to contain the unknown target operator $\boldsymbol{U}_{k^*}^\dagger \boldsymbol{O} \boldsymbol{U}_{k^*}$. Consequently, they exhibit a higher identifiability with a greater probability of producing non-zero measurement outcomes. On the other hand, the average magnitude of the Schmidt coefficients $\{c_{ji}\}_{i=1}^r$ of a highly entangled state $|\boldsymbol{\psi}_j\rangle$ decreases with the Schmidt rank $r$. This leads to a small probability of the target operator $\boldsymbol{U}_{k^*}^\dagger \boldsymbol{O} \boldsymbol{U}_{k^*}$ being measured according to the Born rule. As a result, achieving a close approximation necessitates a larger number of measurements. For instance, the entangled states with $r = 2^n$ could always yield non-zero measurement outputs with a large number of measurements. Conversely, if the unentangled state $|\boldsymbol{\psi}_j\rangle\langle\boldsymbol{\psi}_j|$ with $r = 1$ is exactly identical to the target operator $\boldsymbol{U}_{k^*}^\dagger \boldsymbol{O} \boldsymbol{U}_{k^*}$, one measurement is sufficient to identify the target index $k^*$. These observations provide an intuitive indication about the transition role of entangled data from the lens of quantum information that the entangled data could contain more information than unentangled data, but at the same time increase the difficulty of information extraction using projective measurement.

## Data availability
The entangled data generated in this study are available at the Github repository.

## Code availability
The code used in this study are available at the Github repository.

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

## Acknowledgements

Y.L. acknowledges support from National Natural Science Foundation of China (Grant Nos. U23A20318 and 62276195). X.Y. acknowledges support from the National Natural Science Foundation of China (Grant Nos. 12175003 and 12361161602), NSAF (Grant No. U2330201).

## Author contributions

The project was conceived by Y.D. and D.T. Theoretical results were proved by X.W., Y.D. and Z.T. Numerical simulations and analysis were performed by X.W., Y.D., Y.L. and X.Y. All authors contributed to the write-up.

## Competing interests

The authors declare no competing interests.
