## [Peer Review File · Nature Communications]

Transition Role of Entangled Data in Quantum Machine LearningREVIEWERS' COMMENTS:

Reviewer #1 (Remarks to the Author):

This work studies the effects of entanglement in the dataset for a specific quantum learning problem. Briefly, given a unitary U and an observable O , the problem is to identify U' such that U and U' are consistent on the observable O . The data of this problem is a set of quantum states and their measurement outcomes after applying U and O . The authors showed that entanglement of the data set (i.e., the quantum states are mixed states) can help to reduce the prediction error when there are sufficiently many measurements. In fact, the authors showed a lower bound on the prediction error in terms of the size of the data set, number of measurements, Schmidt rank (that is related to the degree of entanglement), and the dimension of the unitary. This is a very detailed lower bound on the prediction error of the problem. However, I am unsure how to interpret this result, which I will discuss later.

The proof of theorem 1 relies on generalize Fano's result to the quantum setting. Roughly, the learning task can be characterized by a communication protocol such that the prediction error in the learning task is related to the error probability of the communication game. Then, to bound the error probability, the authors demonstrated lower bounds on the packing number of unitaries with 2ϵ error corresponding to O and upper bounds on the mutual information of the target and the output. Overall, there are some nice techniques for analyzing these quantities.

The authors also provide some experiments to demonstrate their findings.

On the other hand, I am unsure about the authors' interpretation of the main theorem. The authors claimed that the number of measurements (m) affects the role of entanglement in the prediction error. When there are "sufficiently many" measurements, the increment of the entanglement could benefit the prediction error. However, each measurement requires a fresh new state. This actually increases the data size. Then, if we consider the number of measurements as part of the data size, then it looks like entanglement does not provide any advantage to the prediction error. (This follows from the fact that increasing the

entanglement will increase the size of the data set to achieve the same level of prediction error.) From this point of view, I am unsure whether this result will be interesting enough for quantum machine learning.

Overall, this paper is well-written and provides a detailed characterization of the prediction error and the degree of entanglement. The proof of the main theorem involves some nice analysis which might be useful for other related problems. An experiment is included to support their findings.

Reviewer #2 (Remarks to the Author):

In this work, the authors have extended prior results on quantum no-free-lunch theorem to the incoherent setting, where the goal is to predict the functions of the form $\text{Tr}(O U \psi U^\dagger)$. Prior papers consider the coherent setting where the risk function was defined in terms of the Hilbert-Schmidt distance between the target unitary and the hypothesis. However, in prior papers, the sampling cost was not considered in the analysis of the prediction error. The main results that the authors prove is that under certain conditions, high Schmidt rank (where the entanglement is defined with a reference system) does not lead the low prediction error if the number of shots are small. This result is a bit counterintuitive from prior results and it highlights how different cost functions (incoherent vs coherent) can lead to different conclusions.

There are several subtleties which make the manuscript not suitable for the audience of Nature Comm. I highlight them below.

1. The authors have restricted the observable to be a projective measurement? Do these results not hold for generic measurement? In my opinion, it is a strong limitations of results.
2. In my opinion, it is a bit contrived that the authors need to assume the training error to be $O(1/2^n)$. Is there a way to generalize these results to an arbitrary ϵ ? For a large system, the training is likely to get challenging as the landscape becomes flat (this depends on the hypothesis class but without much information about the target unitary), it likely that training error will be large. Would the results not hold in that setting? In my opinion, the training error going down exponentially with number of qubits is a highly contrived setting. On the other hand, results of Caro et. al (Generalization in quantum machine learning from few training data, [arXiv:2111.05292](https://arxiv.org/abs/2111.05292)) provides guarantees for a good generalization error when the training error is ϵ . Is it the limitations of techniques that the authors demand $\epsilon = O(1/2^n)$ for their proofs?
3. Minor comment 1: The equation in Theorem 1 should be stated as a lower bound instead of an equality.
4. Could the authors clarify briefly in the main text what do the expectations over **all** unitary, states, and o_j imply in Eq. 3?
5. Could the authors write the right-hand side of Eq. 3 for the case of $r=2^n$ explicitly? This would help the readers connect with the previous results on this topic. Currently, Eq. (3) is written with $\tilde{\epsilon}_1$ and $\tilde{\epsilon}_2$ which makes it difficult to read.
6. In prior work, it was shown that the bound is saturated when the input states are linearly independent but not orthonormal. Could the authors write explicitly for which data sets the bound can be saturated?
7. Could the authors clarify what they imply by this sentence: “Taken together, while the entangled data hold the promise of gaining advantages in terms of the prediction error, they may be inferior to the training data without entanglement in terms of the query complexity.”
8. Numerical experiments: what is the reasoning for keeping $U_j^\dagger O U_j$ orthonormal? Was this also an assumption in the theorem statement?
9. I recommend performing large scale simulations as it can improve the quality of the manuscript!
10. Numerical experiments are not presented properly. Do the authors first run the training with the finite number of shots? What is the training error that they achieve? How would

the training look like as the system size increases? This will be major problem as achieving $1/2^n$ error in training is quite a daunting task!

11. In the caption of Fig 2, the authors write $2/d^2$ but d is not defined.
12. Could the authors connect $r > \sqrt{m/c_1 n}$ condition with different plots in Fig 2 b? Particularly for let's say $r=2$, $m = 100$ and $r= 16$ and $m =100$. Currently, it is not obvious why $r=2$ and $m=100$ should perform worse than $r=16$ and $m=100$!

Overall, the results are quite restrictive due to points 1 and 2. Moreover, the authors could improve the presentation style by stating the assumptions more clearly throughout the manuscript.

Response to reviewers for the manuscript “*transition role of entangled data in quantum machine learning*”

We are grateful for the time invested in the referral of our manuscript and for the helpful feedback. We notice that both of the reviewers raised concerns about the broad impact of our work from different considerations. In this regard, we organize this response letter by first re-emphasizing the significance of the studied problem and the impact of our results in the field of quantum machine learning. Subsequently, we separately address the two reviewer’s concerns by presenting a point-wise reply according to the specific comments. All changes appeared in the revised manuscript are highlighted with the orange color.

Reviewer Point P 0.1 — Both of the reviewers raised concerns about the broad impact of our results in the framework of the quantum no-free-lunch theorem from different considerations.

Reply to all reviewers: The main aspects of the broad impacts of our work can be summarized as follows (refer to subsequent replies for more details):

The significance of studying (quantum) no-free-lunch theorem: The No-Free-Lunch (NFL) theorem is a fundamental concept in artificial intelligence (AI) that characterizes the capabilities of learning models. One interpretation of NFL theorem is that

“the size of the training data determines the ultimate performance of models across different data types, regardless of the optimization procedure used.”

The significant role of training data motivates us to explore **the potential of establishing quantum data-centric AI to further advance quantum machine learning (QML)**. One particularly promising avenue is the utilization of entanglement to create entangled datasets that diverse QML models, as **entangled quantum states are capable of storing more information than that of unentangled states**. Moreover, entangled states have been broadly utilized as **critical resources for numerous quantum algorithms with provable speedups**, including many fault-tolerant quantum algorithms [1, 2, 3], quantum sensing and metrology [4, 5], and quantum channel discrimination [6, 7]. In this regard, establishing a quantum NFL theorem could provide an analytical understanding of the effect of entangled data on determining model performance with the following benefits.

1. Quantum NFL theorem is **problem-independent and optimization-independent**. In particular, quantum NFL theorem considers the average performance over a class of learning problems, which distinguishes from prior literature such that the quantum advantages are recognized by focusing on handcraft data [8, 9] or specific problems [6, 7]. In addition, quantum NFL theorem only concerns the training error the optimizer can achieve but not the specific optimizer, which avoids many notorious problems in the optimization of QML models encounters such as barren plateau [10, 11, 12] and convergence [13, 14, 15, 16]. The problem-independent and optimization independent character of quantum NFL **allow us to compare the performance of QML models and classical ML models from the perspective of the resource used in constructing training data**, providing a deeper understanding of potential quantum advantages in general scenarios.

2. **Unlike the classical NFL where the size of training examples is the only resource, quantum NFL theorem characterizes the effect of the multiple resources utilization used in building the training dataset on the learning performance**, including the training data size N , the Schmidt rank r , the number of measurements m , the possible outcomes of used POVM ℓ for collecting response. The relevant results can deepen our understanding about the separation between quantum and classical learning models.

The implications of our work: The studied problem in our manuscript, i.e., learning the target function $f_U(\rho) = \text{Tr}(OU\rho U^\dagger)$, covers **an important class of learning problems [17, 18]**. In the original version, we specify the observable as the projective measurement. Exploring the setting of projective measurement **relates to an important class of learning tasks, classification**. For instance, projective measurements can be used to distinguish between different classes of data. After processing the quantum states representing data points through a quantum circuit U , projective measurements are performed to read out the result, which corresponds to the classification of the data. Besides, learning the target unitary U under projective measurement O is equivalent to the task of quantum state tomography on the pure state $U^\dagger O U$. This equivalence provides a theoretical understanding of the hardness of the learning task in QML from the lens of quantum information (Refer the reply to Point P 2.2 for details).

Additional theoretical analysis for more general measurements: While Reviewer 2 raised concerns about the restriction of considering projective measurement, we have significantly improved our manuscript by **adding theoretical analysis for more general observables and positive operator-valued measurement (POVM)** in the revised version. This setting covers the **shadow-based learning models [17, 19, 20]**, where the expectation value of O is estimated by post-processing the outputs of random measurements. We show that the transition phenomenon of entangled data occurs for any observable and POVM. These results further consolidate our claim:

“entangled data decreases the amount of extracted information from arbitrary single-copy measurement,”

which echoes with the results in quantum state tomography such that the query complexity scales linearly with the rank of mixed states [21] (see Reply to Point P 2.2 for details).

The diversity of the evaluation metrics in QML: While Reviewer 1 concerns the meaning of the achieved results under various complexity metrics, we would like to emphasize that the complexity in quantum learning theory is a multi-faceted concept [22]. In our manuscript, we explore the impact of entangled data on the learning performance under various evaluation metrics, including sample complexity, query complexity, and prediction error. We would like to highlight that **each of them has its practical meaning in evaluating the potential advantages of different QML models [23] and has been extensively studied in previous studies [8, 9, 17, 24]** (see Reply to Point P 1.2 for details).

The significance of the achieved results under various evaluation metrics: While previous works [17, 25, 26] show that incorporating entanglement into operators or measurements could improve the performance of the learning model, the impact of employing entangled data remains elusive. We analytically show that **contrary to previous findings** that entanglement contributes to learning performance, we obtain a novel insight about the role of entanglement in QML that the impact of entangled data on prediction error exhibits a **dual effect**, depending on the number of permitted measurements. This indicates a high-level understanding from the lens of quantum information theory that **entangled data decreases the amount of extracted information from a single projective measurement**.

Reviewer 1

Reviewer Point P 1.1 — Overall, this paper is well-written and provides a detailed characterization of the prediction error and the degree of entanglement. The proof of the main theorem involves some nice analysis which might be useful for other related problems. An experiment is included to support their findings.

Reply: We appreciate the positive affirmation raised by the reviewer.

Reviewer Point P 1.2 — On the other hand, I am unsure about the authors' interpretation of the main theorem. The authors claimed that the number of measurements (m) affects the role of entanglement in the prediction error. When there are "sufficiently many" measurements, the increment of the entanglement could benefit the prediction error. However, each measurement requires a fresh new state. This actually increases the data size. Then, if we consider the number of measurements as part of the data size, then it looks like entanglement does not provide any advantage to the prediction error. (This follows from the fact that increasing the entanglement will increase the size of the data set to achieve the same level of prediction error.) From this point of view, I am unsure whether this result will be interesting enough for quantum machine learning.

Reply: Thanks for the reviewer's comment, we agree with the reviewer's viewpoint '*if we consider the number of measurements as part of the data size, ... increasing the entanglement will increase the size of the data set to achieve the same level of prediction error.*' However, this implication for considering the number of measurements as part of the data size, i.e., query complexity, is also non-trivial and has been discussed in the original manuscript. In the rest of the reply, we elaborate on the broad impact of the achieved result under various evaluation metrics.

The diversity of evaluation metrics in QML: As explained in reply to Point P 0.1, we would like to emphasize that **the complexity of quantum learning is a multi-faceted concept** [23], including query complexity (a.k.a, copy complexity) and sample complexity (a.k.a, data complexity). **Each of these metrics has its practical meaning** and has been extensively studied in prior literature to explore the potential quantum advantages in the field of QML. To ease of understanding, we categorize and summarize the related works according to the employed evaluation metrics in Table 1.

In particular, Ref. [8] studied the generalization of quantum kernels in terms of **sample complexity** and achieved provable quantum advantage for some specific data. Refs. [24, 27, 30] studied the generalization of quantum learning models in terms of **sample complexity** from various aspects, like data encoding, data distribution, and circuit complexity. Moreover, Refs. [17, 25, 29] studied the **query complexity** of learning quantum processes with and without quantum memory. Besides, instead of considering the query complexity, the sample complexity and the number of measurements for each input state are also considered in many existing works. Refs. [9] and [31] studied the generalization error bound of quantum kernels in terms of sample complexity and the number of measurements under the setting of finite measurements. It is meaningful to separately consider the sample complexity and the number of measurements in the sense that some training states are easy to prepare many times while preparing a large set of different training states is hard in many practical scenarios. This means that given the same query complexity, obtaining the training dataset consisting of a large number of copies of a few easily prepared states is easier than preparing a lot of distinct states with a few copies.

The significance of the achieved results: We would like to highlight that the achieved results in terms of both sample complexity and query complexity are non-trivial in QML, as explained in the reply

Sample complexity	
Description	Related work
The amount of training examples determines the achievable accuracy for data-driven methods. Sample complexity, denoted as N , refers to the requirements in terms of the number of distinct input states on the performance of a learning algorithm.	Nat Commun 12, 2631 [8]: studying the generalization of quantum kernels; Nat Commun 13, 4919 [24]: studying the generalization of VQAs with finite parameterized gats; Nat Commun 14, 3751 [27]: studying the out-of-distribution generalization in learning quantum unitary; Phys. Rev. Lett. 128, 070501 [28]: studying quantum NFL theorem.
Query complexity	
Description	Related work
With quantum data, the number of copies available of a given quantum state, i.e., the number of measurements m , determines the amount of information that can be extracted from the state. Query complexity, denoted as Q , reflects the requirements in terms of copies of all quantum states in the training dataset that are needed to ensure given accuracy levels. In this regard, we have $Q = N \cdot m$.	Phys. Rev. Lett. 126, 190505 [17], and Science 376, 1182-1186 [25], and FOCS52979.2021.00063 [29]: studying the quantum advantage of learning quantum processes with quantum memory.

Table 1: Definitions of sample complexity and query complexity, and the related work in the field of QML.

to Point P0.1. Moreover, we would like to kindly remind the reviewer that **the discussion of the connection between prediction error and query complexity has already been presented in our original manuscript:**

1. (Line 227, Page 3, original main text) “*The results in Theorem 1 indicate that the minimum query complexity is $Nm = \Omega(4^n r c_1 c_2)$, implying the proportional relation between the entanglement degree r and the query complexity.*”
2. (Line 250, Page 3, original main text) “*Taken together, while the entangled data hold the promise of gaining advantages in terms of the sample complexity for achieving the same level of prediction error, they may be inferior to the training data without entanglement in terms of query complexity.*”

These discussions indicate that the results the reviewer mentioned ‘*Increasing the entanglement will increase the size of the data set to achieve the same level of prediction error*’ are also **non-trivial**. Particularly, previous works show that the sample complexity and query complexity can benefit from using entanglement in quantum data or quantum measurements [17, 25, 28]. Additionally, in the task

of quantum channel discrimination, entangled states have been shown to allow for correct discrimination with a strictly higher probability than unentangled states [6, 7]. In contrast to previous work with a positive attitude towards entangled data, our results **counterintuitively indicate that in learning-based tasks, highly entangled data could lead to a high sample complexity and always results in a high query complexity** for achieving the same level of prediction error. These results provide practical guidance on the utilization of entangled data under various practical considerations of quantum resources.

To address the reviewer’s concern, we have appended the following words in the revised manuscript

1. The elaboration about the diversity of the evaluation metric in quantum machine learning is adapted into the above explanations in the Supplementary Material (SM A, Page 15-16). Particularly, we give detailed introductions about the statistical complexity as well as a review of their utilization in quantum machine learning.
2. (Line 229-233, Page 3, main text) **The achieved results in terms of query complexity are also non-trivial, as previous works show that query complexity can benefit from using entanglement in quantum data [6, 7] and quantum measurements [17, 25].**

Reviewer 2

Reviewer Point P 2.1 — In this work, the authors have extended prior results on quantum no-free-lunch theorem to the incoherent setting, where the goal is to predict the functions of the form $Tr(OU|\psi\rangle\langle\psi|U^\dagger)$. Prior papers consider the coherent setting where the risk function was defined in terms of the Hilbert-Schmidt distance between the target unitary and the hypothesis. However, in prior papers, the sampling cost was not considered in the analysis of the prediction error. The main results that the authors prove is that under certain conditions, high Schmidt rank (where the entanglement is defined with a reference system) does not lead the low prediction error if the number of shots are small. This result is a bit counterintuitive from prior results and it highlights how different cost functions (incoherent vs coherent) can lead to different conclusions.

There are several subtleties which make the manuscript not suitable for the audience of Nature Comm.

Reply: We appreciate the reviewer’s valuable time in giving us helpful comments. However, we respectfully disagree with the reviewer’s comment ‘*This result ... highlights how different cost functions (incoherent vs coherent) can lead to different conclusions*’. In particular, the studied problem in **our work differs from that studied in Ref. [28] in both the learning problem and the formalism of training data**, rather than just in the cost function. For convenience, we summarize the major differences between our work and prior papers as follows.

1. **Different learning problems:** Different from Ref. [28] focusing on predicting the output states evolved by a unitary U , our aim is learning the operator $U^\dagger OU$ to accurately predict the expectation value of the observable O on the output states, which covers a wide class of important learning tasks in QML elucidated in the reply of Point P 0.1.

2. **Different training data:** Different from Ref. [28] that refers to the pairs of input-output quantum states $\{|\psi\rangle, (U \times \mathbb{I})|\psi\rangle\}$, we employ the quantum-classical pairs $\{(|\psi\rangle, \mathbf{o})\}$ as the training data, where \mathbf{o} refers to the vectors of the measurement outcomes on the output state $(U \times \mathbb{I})|\psi\rangle$.

In this context, it is **inaccurate** to summarize that our work is a straightforward extension of Ref. [28] on the cost function, because of the **different learning problems and different training data formalism**. Besides, the consideration of finite measurements leads to **fundamental differences** between our work and Ref. [28] in practical implications and theoretical proof, as stated below.

- **Practical implications:** A critical challenge in leveraging quantum systems is the **efficient extraction of information**, which is notably impeded by finite measurements. Such limitations could lead to **markedly divergent conclusions in realizing potential quantum advantages**, heavily dependent on whether these finite measurements are incorporated into the analysis or not [32].
- **Theoretical techniques:** While the derivation of the quantum NFL theorem in Ref. [28] primarily employs algebraic manipulation of input-output states and unitary operators, our study establishes the quantum NFL theorem from an **information-theoretic standpoint**, which involves the utilization of many complicated techniques such as mutual information and Fano’s inequality.

In response to the concern regarding the alignment of our work with the acceptance criteria of Nature Communications, we would like to re-emphasize our contributions stated in the reply to Point P 0.1.

- We provide a rigorous analysis of **how various resources, especially the entanglement degree of input states, affect the model performance by establishing the quantum NFL theorem**. These resources include the training data size N , the Schmidt rank r of entangled states $|\psi\rangle$, the number of measurements m , and the possible outcomes of POVM ℓ , where the POVM is considered in the revised manuscript as further discussed in subsequent replies. The achieved results are **optimization-independent, providing a deeper understanding of potential quantum advantage from the perspective of quantum resources used in building training datasets**.
- We obtain a **novel and counterintuitive** insight about the role of entanglement in QML that **the impact of entangled data on prediction error exhibits a dual effect for any observable**, depending on the number of permitted measurements, while previous work [17, 25, 28] showed that entanglement contributes to learning performance. These insights could provide critical guidance for designing advanced QML protocols, especially for those tailored for execution on early-stage quantum computers with limited access to quantum resources.
- **We build the connection** between the task of learning $\text{Tr}(OU|\psi\rangle\langle\psi|U^\dagger)$ in QML and the task of **quantum state tomography in quantum information theory**. These connections not only help us to better understand the hardness of the learning task in QML from the perspective of quantum information theory, but also introduce some new techniques for deriving the error bound in QML, while most of the existing literature focuses on the Rademacher complexity [8, 22], VC dimension and metric entropy [24, 33].

Based on these contributions, we believe that our submission delivers a substantial advancement in understanding the power of entanglement in QML. We humbly ask you to reconsider the potential impact and significance of our study.

To address the reviewer’s concerns, we append the following words in the updated manuscript.

2. (Line 97, Page 2, main text) In the incoherent learning scenario, the quantum learner is restricted to utilizing datasets with varying degrees of entanglement to operate on an unknown unitary and inferring its dynamics using the finite measurement outcomes collected under the projective measurement, **differing from Ref. [28] in learning problems and training data.**

2. We adapt the above discussion about the differences between Ref. [28] and our work to the Supplementary Material (i.e., SM B, Page 19)

Reviewer 2: “I highlight them below:”

Reviewer Point P 2.2 — The authors have restricted the observable to be a projective measurement? Do these results not hold for generic measurement? In my opinion, it is a strong limitations of results.

Reply: We sincerely thank the reviewer for the suggestion of extending our results to the generic measurements, which can greatly improve the quality of our work. In this reply, we address the reviewer’s concern from two aspects. First, we elucidate **the significance of exploring projective measurements**. Then, we follow the reviewer’s advice to **generalize the results to the case of generic measurements**. In particular, we **show that the transition role of entangled data still occurs for generic measurement**. It is noteworthy that these achieved results have broad impacts on both the analytical understanding of entanglement in QML and practical guidance for devising learning protocols, as illustrated in the reply to Point P 0.1.

The significance of considering projective measurement: We recall the reply to Point P 0.1 that taking the projective measurement as observable in the learning task $f_U(\rho) = \text{Tr}(OU\rho U^\dagger)$ is **related to the classification task [24, 34, 35]**. In classification tasks related to quantum data, projective measurements can be used to distinguish between different classes of data. Initially, the quantum states representing data points are processed through a parameterized quantum circuit $V(\theta)$. Following this processing, projective measurements are carried out on the resultant quantum states. The outcome of these measurements directly corresponds to the classification category of each data point. On the other hand, learning the target unitary U under projective measurement O is equivalent to the task of quantum state tomography on the pure state $U^\dagger OU$. This equivalence provides a theoretical understanding of the hardness of the learning task in QML from the lens of quantum information.

Additional theoretical analysis for generic measurement: To further address the reviewer’s concern, **we discuss the case for generic measurement from two aspects**, namely, the observable used in the target function $f_U(\rho) = \text{Tr}(U^\dagger OU\rho)$ and the measurement used for collecting the response of training data σ , where U and ρ refer to the target unitary and input states respectively. Taking this consideration is because the responses for given input quantum states can be obtained through measuring the output states with arbitrary measurement strategies, but not necessarily employing the fixed observable O . Representative examples in this context are shadow-based learning algorithms [17, 19, 20], where the expectation value of O is estimated by post-processing the outputs of random measurements.

More precisely, we consider a general learning task of learning the target function $f_U(\rho) = \text{Tr}(U^\dagger OU\rho)$ as discussed in the reply to P 0.1 with $U \in \text{SU}(2^n)$ being the n -qubit target unitary and O being arbitrary fixed observable, while the response for given input states is collected from measuring the output states

with ℓ -outcome POVM. Particularly, the training dataset for learning $f_U(\rho) = \text{Tr}(U^\dagger O U \rho)$ in this case refers to $\mathcal{S}_\ell = \{(|\psi_j\rangle, \mathbf{a}_j) : |\psi_j\rangle \in \mathcal{H}_{\mathcal{X}\mathcal{R}}, \mathbf{a}_j = (\mathbf{a}_{j1}, \dots, \mathbf{a}_{jm}), \mathbf{a}_{jk} \in \{z_1, \dots, z_\ell\}_{j=1}^N\}$, where $|\psi_j\rangle$ refers to the entangled states with Schmidt rank r , \mathbf{o}_j is the m -measurement outputs with ℓ -outcome POVM on the system \mathcal{X} , and $\{z_i\}_{i=1}^\ell$ is the ℓ possible outcomes of the employed POVM. In this case, denoting the learned unitary as $V_{\mathcal{S}_\ell}$, the lower bound of the averaged prediction error yields

$$\mathbb{E}_U \mathbb{E}_{\mathcal{S}_\ell} R_U(V_{\mathcal{S}_\ell}) \geq \varepsilon^2 \left(1 - \frac{N \cdot \min\{4m/r, 6m\ell/r(d-1), r \log(d)\} + \log 2}{\log(|\mathcal{X}_{2\varepsilon}(O)|)} \right). \quad (1)$$

where $d = 2^n$ refers to the dimension of n -qubit system, $|\mathcal{X}_{2\varepsilon}(O)|$ refers to the cardinality of the 2ε -packing of the target function space $\mathcal{F} = \{f_U(\rho) = \text{Tr}(U^\dagger O U \rho) : U \in \mathbb{S}\mathbb{U}(2^n)\}$, which only depends on ε and the employed observable O , and is independent on the interested quantities like Schmidt rank r , the number of measurements m , and the training dataset size N . Hence, the term $\log(|\mathcal{X}_{2\varepsilon}(O)|)$ does not affect the discussion of the impact of entangled data on prediction error. Comparing the derived lower bounds under ℓ -outcome POVM with that under projective measurement in the original manuscript given by

$$\mathbb{E}_U \mathbb{E}_{\mathcal{S}} R_U(V_{\mathcal{S}}) \geq \varepsilon^2 \left(1 - \frac{N \cdot \min\{4m/c_1 r d, r \log(d)\} + \log 2}{\log(|\mathcal{X}_{2\varepsilon}(O)|)} \right), \quad (2)$$

we have the following implications:

- **The transition role of entangled data still holds for more generic measurements.** In particular, no matter how large the number of possible outcomes of POVM ℓ is, increasing the Schmidt rank will decrease the prediction error as long as the number of measurements m satisfies $\min\{4m/r, 6m\ell/r(d-1)\} \leq r \log(d)$, and increase the prediction error otherwise.
- **Increasing the number of possible outcomes of POVM ℓ can exponentially reduce the number of measurements** required to achieve the same level of prediction error. Particularly, when the number of measurements m and the number of outcomes ℓ satisfies $4m/r \leq 6m\ell/r(d-1) \leq r \log(d)$, then the minimum term in Eqn. (1) and in Eqn. (2) refer to $4m/r$ and $4m/c_1 r d$, where the former term related to the case of POVM removes a factor $1/d = 1/2^n$ of the latter term related to the case of projective measurement.
- When the number of possible outcomes ℓ is of constant order and hence $6m\ell/r(d-1) \leq 4m/r$, the achieved result in Eqn. (1) reduces to the results achieved in Eqn. (2) for the case of employing projective measurement up to a constant factor.

Conclusively, the achieved results indicate that **the transition role of entangled data still holds** for arbitrary observable and arbitrary ℓ -outcome POVM, and **increasing the possible outcomes of POVM ℓ could significantly reduce the required query complexity** for achieving the same level of prediction error.

Connection with results in quantum state tomography: The achieved results for generic measurements can be intuitively understood from the perspective of quantum state tomography. Differing from the connection discussed above for projective measurement where the operator $U^\dagger O U$ refers to a pure state, here we discuss the connection of learning the operator $U^\dagger O U$ for generic measurement with quantum state tomography on the output states $U \rho_j U^\dagger$. Particularly, obtaining the information of the n -qubit target unitary under given observable O requires employing sufficient many input states $\{|\psi_j\rangle\}_{j=1}^N$

(known as probe states in the context of quantum channel discrimination) and precise estimation of the expectation value $\text{Tr}((U^\dagger O U \otimes \mathbb{I}) |\psi_j\rangle \langle \psi_j|)$ [36]. While the requirement of training data size N could be alleviated by utilizing highly entangled states, obtaining an approximate statistical estimation of the expectation value of O requires a sufficient number of measurements. For general observable O , an effective approach is to perform tomography on the output state $U^\dagger \rho U$ with $\rho_j = \text{Tr}_{\mathcal{R}}(|\psi_j\rangle \langle \psi_j|)$ being the partial traced states. The number of measurements, or equivalently the query complexity, for state tomography of rank r mixed states with ℓ -outcome POVM have been well studied with obtaining a tight bound of $\Theta(4^{nr^2}/\ell\varepsilon^2)$ [21], where the rank r of mixed state ρ_j on the quantum system \mathcal{X} corresponds to the Schmidt rank of entangled state $|\psi_j\rangle$ on the quantum system \mathcal{XR} . This accords with our results that **extracting information from highly entangled data requires more measurements and increasing the number of possible outcomes of employed POVM ℓ can significantly reduce the number of measurements**, as shown by the term $6m\ell/r(d-1)$ in Eqn. (1).

The revised words with respect to the significance of taking a projective measurement as observable and the results for generic measurement are summarized as follows.

1. (Line 113-118, Page 2, main text) **Moreover, we generalize the results to the case of ℓ -outcome positive-operator valued measure (POVM). We show that the transition role still holds for arbitrary POVM and increasing the possible outcomes ℓ could significantly reduce the query complexity.**
2. We provide an explanation of the significance of the learning problem with taking a projective measurement as observable in Supplementary Material (i.e., SM A, Page 16).
3. The results for generic measurements are summarized in Theorem 2 in the main text. The elaboration of the related proof is also adapted to the updated manuscript, where we append a new section entitled by 'The lower bound for using POVM' in Supplementary Material (i.e., SM D, Page 32-36)

Reviewer Point P 2.3 — In my opinion, it is a bit contrived that the authors need to assume the training error to be $O(1/2^n)$. Is there a way to generalize these results to an arbitrary ϵ ? For a large system, the training is likely to get challenging as the landscape becomes flat (this depends on the hypothesis class but without much information about the target unitary), it likely that training error will be large. Would the results not hold in that setting? In my opinion, the training error going down exponentially with number of qubits is a highly contrived setting. On the other hand, results of Caro et. al (Generalization in quantum machine learning from few training data, arXiv:2111.05292) provides guarantees for a good generalization error when the training error is ϵ . Is it the limitations of techniques that the authors demand $\epsilon = O(1/2^n)$ for their proofs?

Reply: We appreciate the reviewer's comment but respectfully disagree with the viewpoint "*the scaling of training error $\epsilon = O(1/2^n)$ is a restrictive assumption*". To better address the reviewer's concerns, in the following, we would like to first highlight that **the explored topics of our manuscript and Caro et. al [24] are fundamentally different**. Then we will elaborate on that **the scaling of training error $\epsilon = O(1/2^n)$ is ground truth but not a restrictive assumption**.

The difference from the studied problem in Ref. [24]: As we illustrated in the reply to Point P 0.1, the study of quantum learning theory is a diverse field. We summarize the difference between the studied problem in Ref. [24] and that in our manuscript as follows.

- Quantum NLF is **problem-independent** but Ref. [24] focuses on the **problem-dependent risk**. In particular, the risk function defined in Ref. [24] and our study respectively refer to

$$\text{Ours: } \mathbb{E}_U \mathbb{E}_S \left(\text{Tr} \left(OU \rho U^\dagger \right) - \text{Tr} \left(OV_S \rho V_S^\dagger \right) \right)^2, \quad (3)$$

$$\text{Adapted from Ref. [24]: } \mathbb{E}_{(x,y) \sim \mathcal{P}} \left(y - \text{Tr} \left(OV(\theta) \rho(x) V(\theta)^\dagger \right) \right)^2, \quad (4)$$

where Ref. [24] considers learning from the training data (x, y) following specific distribution \mathcal{P} with specific parameterized quantum circuits (PQC) $V(\theta)$ consisting of finite gate parameters θ . In this case, the whole PQC could not form a t -design. However, for quantum NFL theorem, it concerns the task of learning an arbitrary Haar unitary which could consist of exponential quantum gates, and the training dataset \mathcal{S} being uniformly sampled from all possible states.

- Ref. [24] considers the learning problem in the setting of **an infinite number of measurements**, while we consider the setting of a finite number of measurements and the response being collected from arbitrary ℓ -outcome POVM.
- Ref. [24] investigates the **upper bound** of the generalization error in terms of the training data size N and the number of parameterized gates T , while we study the **lower bound** of the prediction error in terms of the Schmidt rank r , the training data size N , and the number of measurement m .

The misunderstanding about the training error scaling: Taking the above differences into consideration, we would like to emphasize that $\varepsilon = \mathcal{O}(1/2^n)$ **is not an assumption but a ground truth** when considering the projective measurement and the framework of quantum NFL theorem.

Particularly, quantum NFL theorem considers **the average performance, including the training error** of the quantum learning models **over Haar unitaries**. Considering the Haar random unitary U like the risk defined in Eqn. (3), the expectation value of an arbitrary observable O on the evolved state $U|\psi\rangle$ yields

$$\mathbb{E}_{U \sim \text{Haar}} \text{Tr}(OU\rho U^\dagger) = \frac{\text{Tr}(O)}{2^n} = \frac{\text{Tr}(|o\rangle\langle o|)}{2^n} = \frac{1}{2^n}. \quad (5)$$

This indicates that the scaling of measurement output depends on the employed observable O . A large trace $\text{Tr}(O)$ leads to a large tolerance of training error. On the other hand, the observable is set as the projective measurement $O = |o\rangle\langle o|$ in our original manuscript, leading to the output scaling of $\mathcal{O}(1/2^n)$. In this context, the training error scales naturally as $\mathcal{O}(1/2^n)$, which is easy to reach during training.

To address the reviewer's concern, we append the following words in the revised manuscript.

1. (Lines 262-266, Page 3, main text) *Remark.* (i) The scaling $1/2^n$ in the training error ε and the factor of the achieved lower bound $\tilde{\varepsilon}^2/4^n$ comes from the consideration of average performance over Haar unitaries where the expectation value of observable O scales as $\text{Tr}(O)/2^n$.

2. The elaboration about the scaling of training error and prediction error is adapted to the updated manuscript, where we append a new subsection entitled by ‘The scaling of training error and prediction error’ in Supplementary Material (i.e., SM C, Page 31).

Reviewer Point P 2.4 — Minor comment 1: The equation in Theorem 1 should be stated as a lower bound instead of an equality.

Reply: We have followed the reviewer’s suggestion to employ an inequality to state the lower bound. In the updated version, we have revised the statement of Theorem 1 as follows.

1. **Theorem 1** (Quantum NFL theorem in learning quantum dynamics, informal). *Following the settings in Eqn. (1), suppose that the training error of the learned hypothesis on the training data \mathcal{S} is less than $\varepsilon = \mathcal{O}(1/2^n)$. Then the lower bound of the averaged prediction error in Eqn. (2) yields*

$$\mathbb{E}_{U, \mathcal{S}} R_U(V_S) \geq \Omega \left(\frac{\tilde{\varepsilon}^2}{4^n} \left(1 - \frac{N \cdot \min\{m/(2^n r c_1), rn\}}{2^n c_2} \right) \right),$$

where $c_1 = 128/\tilde{\varepsilon}^2$, $c_2 = \min\{(1 - 2\tilde{\varepsilon})^2, (64\tilde{\varepsilon}^2 - 1)^2\}$, $\tilde{\varepsilon} = \Theta(2^n \varepsilon)$, and the expectation is taken over all target unitary U , entangled states $|\psi_j\rangle$ and measurement outputs o_j .

Reviewer Point P 2.5 — Could the authors clarify briefly in the main text what do the expectations over all unitary, states, and o_j imply in Eq. (3)?

Reply: Thanks for the reviewer’s comment. Here we follow the explanation of reply to Point P 2.3 to address the concerns about the meaning of the expectation over all unitaries, states, and o_j in Eq. (3), i.e.,

$$\mathbb{E}_U \mathbb{E}_{\mathcal{S}} R_U(V_S) = \mathbb{E}_U \mathbb{E}_{\mathcal{S}} \int d\psi (f_U(\psi) - h_S(\psi))^2. \quad (6)$$

Particularly, under the framework of the NFL theorem, it considers the averaged risk over a specified class of target functions and training datasets. Moreover, in the context of QML, the quantum NFL theorem specifies the target function and training dataset as the target unitary and quantum states. Here, we follow the treatments in Ref. [28, 37] choosing the Haar unitary as the target unitary. Additionally, we construct a sampling rule of the input states which approximates the uniform distribution of entangled states with Schmidt rank r . Another difference is that the response for given input states used in our work refers to the random measurement outcomes rather than the deterministic quantum states used in Ref. [28, 37]. In this regard, we have revised the manuscript and added the following sentence to provide additional context and clarity for the reader.

(Lines 161-166, Page 3, main text) *Moreover, we follow the treatments in Ref. [28] choosing the Haar unitary as the target unitary. Additionally, we construct a sampling rule of the training input states which approximates the uniform distribution in the set*

of all entangled states with Schmidt rank r (Please refer to SM B)."

Reviewer Point P 2.6 — Could the authors write the right-hand side of Eq. (3) for the case of $r = 2^n$ explicitly? This would help the readers connect with the previous results on this topic. Currently, Eq. (3) is written with $\tilde{\epsilon}_1$ and $\tilde{\epsilon}_2$ which makes it difficult to read.

Reply: Thanks for the comments. We first recall that the Eqn. (3) the reviewer mentioned should be the equation in Theorem 1, i.e.,

$$\mathbb{E}_{U,S}RU(V_S) \geq \Omega \left(\frac{\tilde{\epsilon}^2}{4^n} \left(1 - \frac{N \cdot \min\{m/(rc_1), rn\}}{2^n c_2} \right) \right)$$

where the left-hand side $\mathbb{E}_{U,S}RU(V_S)$ refers to the averaged risk function in the incoherent learning protocol of with finite measurements, $c_1 = 128(2^n + 1)/\tilde{\epsilon}^2$, $c_2 = \min\{(1 - 2\tilde{\epsilon})^2, (64\tilde{\epsilon}^2 - 1)^2\}$. In the following, we separately address the reviewer's concerns.

Currently, Eq. (3) is written with $\tilde{\epsilon}_1$ and $\tilde{\epsilon}_2$ which makes it difficult to read: If we understand correctly, the term $\tilde{\epsilon}_1$ and $\tilde{\epsilon}_2$ should refer to c_1 and c_2 as there are no terms $\tilde{\epsilon}_1$ and $\tilde{\epsilon}_2$ in Eq. (3). To address the reviewer's concern, we have explicitly written the term 2^n of c_1 in the right-hand side of Eq. (3). In the updated version, the terms c_1 and c_2 only depend on the term $\tilde{\epsilon}$, which is of constant order.

Could the authors write the right-hand side of Eq. (3) for the case of $r = 2^n$ explicitly: The form of the achieved r -dependent lower bound in our manuscript follows the convention of previous work, i.e., Ref. [28], where the achieved lower bound is also r -dependent as given by Eqn. (5) in Ref. [28]

$$\mathbb{E}_{U,S_Q}RU(V_{S_Q}) \geq 1 - \frac{r^2 N^2 + d + 1}{d(d + 1)},$$

where $d = 2^n$ refers to the dimension of n -qubit quantum system, the term in the left-hand side $\mathbb{E}_{U,S_Q}RU(V_{S_Q})$ refers to the averaged risk function in the ideal learning coherent protocol with infinite measurements. Moreover, we would like to kindly remind the reviewer that we have discussed the case of $r = 2^n$ and the connection with previous results in our original manuscript.

1. In Line 180 Page 3, we state that "Accordingly, in the two extreme cases of $r = 1$ and $r = 2^n$, achieving zero averaged risk requires $N = 2^n c_2/n$ and $N = 1$ training input states, where the latter achieves an exponential reduction in the number of training data compared with the former."
2. In Line 184 Page 3, we state that "This observation implies that the entangled data empower QML with provable quantum advantage, which accords with the achieved results of Ref. [50] in the ideal coherent learning protocol with infinite measurements."

To address the reviewer's concern, we have improved our manuscripts and revised the statement of Theorem 1 and the discussion about the case of $r = 2^n$.

1. **Theorem 1** (Quantum NFL theorem in learning quantum dynamics, informal). *Following the settings in Eqn. (1), suppose that the training error of the learned hypothesis on the training data \mathcal{S} is less than $\epsilon = \mathcal{O}(1/2^n)$. Then the lower bound of the averaged prediction error in Eqn. (2) yields*

$$\mathbb{E}_{U,S}RU(V_S) \geq \Omega \left(\frac{\tilde{\epsilon}^2}{4^n} \left(1 - \frac{N \cdot \min\{m/(2^n r c_1), rn\}}{2^n c_2} \right) \right),$$

where $c_1 = 128/\tilde{\varepsilon}^2$, $c_2 = \min\{(1 - 2\tilde{\varepsilon})^2, (64\tilde{\varepsilon}^2 - 1)^2\}$, $\tilde{\varepsilon} = \Theta(2^n \varepsilon)$, and the expectation is taken over all target unitary U , entangled states $|\psi_j\rangle$ and measurement outputs \mathbf{o}_j .

- Line 184, Page 3, main text, we state that “Particularly, when a sufficient number of measurements m is allowed such that the Schmidt rank r obeys $r < \sqrt{m/(c_1 2^n n)}$, the minimum term in the achieved lower bound refers to Nrn and hence increasing r can constantly decrease the prediction error. Accordingly, in the two extreme cases of $r = 1$ and $r = 2^n$, ...”

Reviewer Point P 2.7 — In prior work, it was shown that the bound is saturated when the input states are linearly independent but not orthonormal. Could the authors write explicitly for which data sets the bound can be saturated?

Reply: Thanks for the reviewer’s comment. To address the reviewer’s concern, we first briefly review **the differences between the learning problem studied in previous work [28] and that in our manuscript**. Then we elaborate on why independence and orthogonality of training states should be considered in coherent learning but make less sense in the incoherent learning protocol. Finally, we explain the relation between the training data and the achieved lower bound in the coherent learning protocol.

The difference between the learning setting in Ref. [28] and our manuscript. Recall the reply to Point P 1.1, we have explained the difference between Ref. [28] and our manuscript in the learning problem and training data. Particularly, previous work focuses on learning a target unitary coherently without considering the finite measurement error, which allows access to the output states and leads to the training examples of input-output state pairs $\{(|\psi_j\rangle, U|\psi_j\rangle)\}_{j=1}^N$. This leads to the loss function $\sum_{j=1}^N \|U|\psi_j\rangle\langle\psi_j|U^\dagger - V_Q|\psi_j\rangle\langle\psi_j|V_Q^\dagger\|_1^2$ with V being the learned hypothesis. Moreover, the setting of an infinite number of measurements allows the perfect training assumption $U|\psi_j\rangle = e^{-i\theta_j}V_Q|\psi_j\rangle$ (or equivalently $|\langle\psi_j|V_Q^\dagger U|\psi_j\rangle|^2 = 1$) with θ_j being an unknown global phase. However, in this manuscript, we consider learning the target function $f_U(\rho) = \text{Tr}(OU\rho U^\dagger)$ incoherently with considering the finite measurement error.

The role of the type of training states in various learning settings. We would like to state that **the discussion of the input states being orthogonal or linearly independent is necessary for coherent learning but not for incoherent learning settings, as they highlight the importance of the global phase in saturating the lower bound which can not be captured in incoherent learning settings due to the measurement operation**. Particularly, we first briefly review the main equation (Eqn (A15)—(A18) in Ref. [28]) about the derivation of the lower bound, which is given by

$$\begin{aligned} \int dU |\text{Tr}[U^\dagger V_{S_Q}]|^2 &= \int dY \left| r \left(\sum_{j=1}^t e^{i\theta_j} \right) + \text{Tr}[Y] \right|^2 \\ &= r^2 \left(\left| \sum_{j=1}^t e^{i\theta_j} \right|^2 \right) + \int dY |\text{Tr}[Y]|^2 + \int dY 2r \Re \left[\left(\sum_{j=1}^t e^{i\theta_j} \right) \text{Tr}[Y] \right] \end{aligned}$$

$$\begin{aligned} &\leq r^2 t^2 + \int dY |\text{Tr}[Y]|^2 + \int dY 2r \Re \left[\left(\sum_{j=1}^t e^{i\theta_j} \right) \text{Tr}[Y] \right] \\ &= r^2 t^2 + 1. \end{aligned}$$

The discussion about the orthogonality of input states occurs in the condition enabling the equality to hold in the third line. Moreover, this equality holds when $\theta_1 = \theta_2 = \dots = \theta_t$ with t being the training data size, which requires the input states to be linearly independent but non-orthogonal. Particularly, we notice that the perfect training assumption in Ref. [28] implies that

$$\langle \psi_k | \psi_j \rangle = \langle \psi_k | U U^\dagger | \psi_j \rangle = e^{i(\theta_j - \theta_k)} \langle \psi_k | V_Q V_Q^\dagger | \psi_j \rangle = e^{i(\theta_j - \theta_k)} \langle \psi_k | \psi_j \rangle, \quad (7)$$

where $\theta_j = \theta_k$ for $k \neq j$ holds if and only if $\langle \psi_k | \psi_j \rangle \neq 0$ or equivalently the training states are not orthogonal. Conclusively, the discussion on orthogonality and independence of training states in Ref. [28] is because **the saturation of the achieved lower bound relies on the alignment of the global phase, i.e., $\theta_j = \theta_k$ for all $j, k \in [N]$, which is enabled by the independent but non-orthogonal training states.** However, our results are built on the incoherent learning setting where the output is measurement outcomes but not quantum states and does not depend on the assumption of perfect training. In this regard, the condition of the training states being orthogonal or non-orthogonal would not affect the saturation of the lower bound.

The relation between the type of training states and the achieved lower bound. While the proof techniques in [28] only involve linear algebra, our manuscript employs more intricate approaches, involving the utilization of Fano's inequality and inequalities related to mutual information. It is important to note that throughout our deduction process, the discussion of data types becomes pertinent solely in the context of deriving the mutual information bound between the target unitary and the evolved training states, i.e.,

$$I(U_X, U_X | \psi_1), \dots, U_X | \psi_N) \leq \sum_{j=1}^N I(U_X, U_X | \psi_j), \quad (8)$$

where the inequality follows the conditional subadditivity of mutual information. We note that the equality holds if and only if the training states are linearly independent. However, the saturation of Eqn. (8) can not induce the saturation of the achieved lower bound regarding the prediction error, which involves other scaling processes during the whole derivation.

The tightness of the achieved lower bound. While showing the tightness of each scaling process is complicated, we have shown the tightness of the achieved lower bound in the original manuscript by building the connection with the achieved results in the field of quantum state tomography (Please refer to SM C). In particular, the task of learning unitary U under the projective measurement $O = |o\rangle \langle o|$ is equivalent to quantum state tomography of the pure state $U |o\rangle$, where the query sample complexity has been well studied in prior works [21]. We recall that the statement about the tightness of the achieved lower bound in the original manuscript is as follows

- In Line 1472, Page 30, SM C, we state that 'This implies that the achieved query complexity in Theorem 1 $Nm = \Omega(d^2/\tilde{\epsilon}^2)$ is tight and optimal for quantum state tomography under the non-adaptive measurement with a constant number of outcomes.'

Reviewer Point P 2.8 — Could the authors clarify what they imply by this sentence: “Taken together, while the entangled data hold the promise of gaining advantages in terms of the prediction error, they may be inferior to the training data without entanglement in terms of the query complexity.”

Reply: We thank the reviewer for pointing out this unclear clarification. Before moving to clarify the implication of this sentence, we first recall the definition of sample complexity and query complexity, as well as the related implications of Theorem 1. As defined in Table 1, the sample complexity and query complexity respectively refer to the number of distinct input states and the number of all state copies in the training dataset required to achieve a certain level of accuracy. Particularly, Theorem 1 indicates that given the same sample complexity, the highly entangled data can achieve better prediction error over lowly entangled data if the number of measurements is large enough. However, when considering the same limitation of query complexity (the product of sample complexity and the number of measurements), the entangled data will present a worse performance in prediction error than unentangled data. In summary, we would like to emphasize by this sentence that Theorem 1 delivers various implications from different considerations of complexity metric.

To clarify this implication clearly, we have revised the original sentence in the updated version.

1. (Line 247-251, Page 3, main text) Taken together, while the entangled data hold the promise of gaining advantages in terms of **the sample complexity for achieving the same level of prediction error, they may be inferior to the training data without entanglement in terms of the query complexity.**

Reviewer Point P 2.9 — Numerical experiments: what is the reasoning for keeping $U_j^\dagger O U_j$ orthonormal? Was this also an assumption in the theorem statement?

Reply: Thanks for the reviewer’s comments. We recall that one of the main ideas in the proof of Theorem 1 is to discretize the hypothesis space with 2ε -packing. Then the learning problem could be reduced to the hypothesis testing problem on this discretized hypothesis space. **The function in the discretized hypothesis spaces should be well distinguished under the ϱ -metric.** The setting of keeping operators $U_j^\dagger O U_j$ orthogonal is **not an assumption in the theorem statement, but a specific case of the well-distinguished discrete function spaces.** Particularly, for any two orthogonal operators $U_i^\dagger O U_i$ and $U_j^\dagger O U_j$, the distance of U_i and U_j under ρ -metric refers to

$$\varrho(U_i, U_j) := \frac{1}{\sqrt{2d(d+1)}} \|U_i^\dagger O U_i - U_j^\dagger O U_j\|_1 = \frac{1}{\sqrt{2d(d+1)}}, \quad (9)$$

which reaches the maximal distance under the ϱ -metric. This enables good distinguishability between different unitaries in the 2ε -packing. Additionally, **any discretized function class \mathcal{F} with $\rho(U_i, U_j) \geq \varepsilon$ for any $U_i, U_j \in \mathcal{F}$ could be chosen as the set of target unitaries.** A small ε leads to worse distinguishability, and hence a large number of measurements is required for correctly estimating the target unitary.

To address the reviewer’s concern, we append the following words in the updated manuscript.

1. (Line 329-333, Page 4, main text) The target unitary U_X is chosen uniformly from a discrete set $\{U_i\}_{i=1}^M$, where $M = 2^n$ refers to the set size and the operators $U_j^\dagger O U_j$ with U_j in this set are orthogonal such that the operators $U_j^\dagger O U_j$ are well distinguished.
2. (Line 1701-1704, Page 35, SM E) *Remark.* The construction of orthogonal operators $U^\dagger O U$ leads to the best distinguishability. Additionally, any discretized function class \mathcal{F} with $\rho(U_i, U_j) \geq \varepsilon$ for any $U_i, U_j \in \mathcal{F}$ could be chosen as the set of target unitaries. A small ε leads to worse distinguishability, and hence a large number of measurements is required for correctly estimating the target unitary.

Reviewer Point P 2.10 — I recommend performing large scale simulations as it can improve the quality of the manuscript!

Reply: We have followed the reviewer’s suggestion to perform large-scale simulations to show the impact of entangled data on prediction performance under various limitations of measurement times. Particularly, in the latest version, we conduct numerical simulations for the number of qubits $n = 6$ and $n = 8$, as a larger-scale simulation for the case of $n = 10$ is highly time-consuming. The required time for the numeric simulation of $n = 10$ is listed in Table 2. If following the convention in other qubits setting with running 40 various seeds, the total required time for $n = 10$ is more than 580 days.

Data size (N)	2^0	2^1	2^2	2^3	2^4	2^5	2^6	2^7	2^8	2^9	2^{10}
Times (days)	0.7	1.4	2.6	5.8	14.6	30.9	40.4	71.2	84.8	118.7	189.1

Table 2: **The required running time of learning one 10-qubit unitary with various training data sizes for running 40 seeds.**

We now introduce the setting of other hyperparameters. The Schmit rank r and the training data size N are set as $\{2^0, 2^1, \dots, 2^6\}$ and $\{2^0, 2^1, \dots, 2^8\}$ for the case of $n = 6$ and $n = 8$ respectively. The number of measurements for $n = 6$ and $n = 8$ is set as

$$\begin{aligned}
 & m \in \{10, 100, 500, 1000, 5000, 10000, 20000\} \\
 \text{and } & m \in \{10, 100, 500, 1000, 5000, 10000, 20000, 50000, 100000\}.
 \end{aligned} \tag{10}$$

The simulation results for $n = 6$ and $n = 8$ are depicted in Fig. 1 and Fig. 2, respectively.

The simulation results show that **the transition role of entangled data still occurs for large quantum systems**. Particularly, for a small number of measurements, increasing the Schmidt rank r of entangled data could first decrease the prediction error, and then increase the prediction error once r surpasses some critical point. When the number of measurements m is sufficiently large, increasing the Schmidt rank r constantly decreases the prediction error. For instance, we can see from the results of 8-qubit and training data size $N = 2^7$ that for a small number of measurements $m = 10$, increasing the Schmidt rank r will first decrease the prediction error in the regime of $r \leq \sqrt{m/c_1 n}$ and then increase the prediction error once the Schmidt rank r surpass a critical point such that $r \geq \sqrt{m/c_1 n}$. These phenomenons accord with the theoretical results of Theorem 1 in the original manuscript.

To address the reviewer’s concern, we append the following words to the revised manuscript

The above elaborations are appended to Supplementary Material, i.e., SM E.

Figure 1: **Simulation results of quantum NFL theorem for 6-qubits.** The averaged prediction error with a varied number of measurements m and Schmidt rank r when $N = 2$, $N = 2^3$, and $N = 2^5$. The z-axis refers to the averaged prediction error defined in Eqn. (1). The label ‘ $(\times 2/4^n)$ ’ refers that the plotted prediction error is normalized by a multiplier factor $2/4^n$.

Figure 2: **Simulation results of quantum NFL theorem for 8-qubits.** The notations are identical to those in Fig. 2.

Reviewer Point P 2.11 — Numerical experiments are not presented properly. Do the authors first run the training with the finite number of shots? What is the training error that they achieve? How would the training look like as the system size increases? This will be major problem as achieving $1/2^n$ error in training is quite a daunting task!

Reply: Thanks for the reviewer’s comments. We recall that in the reply to Point P 2.3, we have analytically shown the scaling of training error $\mathcal{O}(1/2^n)$ is a ground truth and easy to reach during training. In this reply, to further address this concern numerically, we will first explain the training process in our numerical simulations, and then present the numerical results of the training error with varying system sizes.

The learning process in our numerical simulations: Before elucidating, we first recall that **the aim of numeric is to verify the relation between the prediction error and the resource used for collecting the training data**, including the training data size N , the Schmidt rank r of the entangled state,

and the number of measurements m . **We do not consider the measurement cost during the training process.** In the numerics, we consider a discrete function space of target concepts $\mathcal{F} = \{f_U(\rho) = \text{Tr}(U_k^\dagger O U_k \rho) : U_k^\dagger O U_k = |e_k\rangle\langle e_k|\}_{k=1}^{2^n}$ where $|e_k\rangle$ refers to the computational basis with k -th entry being 1 and n is the number of qubits. In this case, the operator $U^\dagger O U$ in the function space can be distinguished well (as discussed in the reply to Point P2.9). We uniformly sampled an index $k^* \in \{1, 2, \dots, 2^n\}$ and take $U_{k^*}^\dagger O U_{k^*}$ as the target operator to generate the training dataset $\mathcal{S} = \{\rho_j, \mathbf{o}_j\}_{j=1}^N$ of size N where $\mathbf{o}_j = \sum_{i=1}^m \mathbf{o}_{jm}/m$ refers to the response of input states ρ_j with m being the measurement times. The learning process determines the learned hypothesis $h_{\mathcal{S}}$ from the the same function space as target concept \mathcal{F} by minimizing

$$h_{\mathcal{S}} = \arg \min_{h \in \mathcal{F}} \frac{1}{N} \sum_{i=1}^N (\mathbf{o}_j(h) - \mathbf{o}_j)^2, \quad (11)$$

where $\mathbf{o}_j(h)$ refers to the m -measurements response related to the hypothesis $h \in \mathcal{F}$ of input states ρ_j . **The training process refers to the minimization process of Eqn. (11). Hence the training error is given by**

$$\min_{h \in \mathcal{F}} \frac{1}{N} \sum_{i=1}^N (\mathbf{o}_j(h) - \mathbf{o}_j)^2. \quad (12)$$

We would like to emphasize that as we aim to obtain the information-theoretical lower bound, we **only consider the number of measurements used for collecting the response in training data, but not consider the measurement cost during the training process.** In this regard, the simulation results only consider how the prediction error varies with the data size of \mathcal{S} , the number of measurements m used to obtain the response \mathbf{o}_j , and the Schmidt rank of entangled states r .

The numerical results on the training error: To further address the reviewer's concern. We conduct simulation numerics to verify the argument raised in the reply to Point P2.3 that the training error is scaled as $\mathcal{O}(1/2^n)$. Particularly, we record the training error for varying number of qubits $n \in \{4, 5, 6, 7, 8\}$ and varying Schmidt rank $\{2, 4, 8, 16\}$. The training data size is set as $N = 16$ and the number of measurements is set as $m \in \{10, 100, 1000\}$. The numerical results are presented in Fig. 3, where the training error decreases with scaling 2^n as the system size n increases. This indicates that the training error of scaling $\mathcal{O}(1/2^n)$ is easy to reach during the training process.

To address the reviewer's concern, we append the following words in the revised manuscript.

1. (Line 1723, Page 36, SM E) **This minimization of Eqn. (E2) refers to the training process**, which can be accomplished through direct calculation.
2. (Line 1725-1727, Page 36, SM E) **Notably, we only consider the relation between the prediction error and the resource used for collecting the training data, including the training data size N , the Schmidt rank r of the entangled state, and the number of measurements m . We do not consider the measurement cost during the training process.**
3. We have appended the above simulation results of the training error in Supplementary Material (i.e., SM E2, Page 38-40).

Figure 3: **Training error with varying system size.** The left panel, middle panel, and right panel present the training error for the number of measurements $m = 10, 100, 1000$, respectively.

Reviewer Point P 2.12 — In the caption of Fig 2, the authors write $2/d^2$ but d is not defined.

Reply: Thank the reviewer for pointing out this typo. We have revised the notation $2/d^2$ by $2/n^4$ in the manuscript, where n is the number of qubits.

Reviewer Point P 2.13 — Could the authors connect $r > \sqrt{m/c_1 n}$ condition with different plots in Fig 2 b? Particularly for let's say $r=2, m = 100$ and $r = 16$ and $m = 100$. Currently, it is not obvious why $r=2$ and $m=100$ should perform worse than $r=16$ and $m=100$!

Reply: Thanks for the reviewer's comments. We have followed the reviewer's suggestion to append additional plots to clarify the confusion. Particularly, we would like to state that the case of $r = 2$ & $m = 100$ performing worse than the case of $r = 16$ & $m = 100$ is because the number of measurements being $m = 100$ for both the cases of $r = 2$ and $r = 16$ lies the area of $r \leq \sqrt{m/c_1 n}$ where a large Schmidt rank r leads to a small prediction error. To this end, in the revised version, we replace the setting of $m = 100$ with $m = 10$ in Fig. 2(b) of the original manuscript, such that both the cases of $r = 2$ and $r = 16$ lie the area of $r \geq \sqrt{m/c_1 n}$. The results are shown in Fig. 4 (b), where the case of $r = 2$ & $m = 10$ performs better than the case of $r = 16$ & $m = 10$, according with the theoretical results that highly entangled data could lead to a large prediction error for a small number of measurements.

Moreover, we present an additional plot to show the prediction performance of Schmidt rank taking $r \in \{2, 4, 6, 8, 12, 16\}$ with a varying number of measurements $m \in \{10, 100, 300, 500, 800, 1000, 2000, 5000, 10000\}$. For clearness, in the following, we present the appended results and discuss their implications. As Fig 5 shows, when the number of measurements is small with $m = 10$, the prediction error for the case of $r = 2$ achieves the best performance over other highly entangled data for both cases of training data size being $N = 2$ and $N = 8$. On the other hand, when the number of measurements increases to $m = 100$, the entangled data with $r = 2$ reaches the best prediction performance in itself and keeps invariant with increasing m but has the worst performance than other highly entangled data. This means $m = 100$ with $r = 2$ lies in the region of $r < \sqrt{m/c_1 n}$ where the number of measurements is enough for extracting all information from the entangled states. In this case, increasing the entanglement degree r helps to obtain more information about the target unitary and hence decreases the prediction error. Notably, these numerical results echo with our theoretical results.

Figure 4: **Simulation results of quantum NFL theorem when incoherently learning quantum dynamics.** (a) The averaged prediction error with a varied number of measurements m and Schmidt rank r when $N = 2$ and $N = 8$. The z-axis refers to the averaged prediction error defined in Eqn. (1). (b) The averaged prediction error with the varied sizes of training data. The label ' $r = a \& m = b$ ' refers that the Schmidt rank is a and the number of measurements is b . The label ' $(\times 2/4^N)$ ' refers that the plotted prediction error is normalized by a multiplier factor $2/4^N$.

To address the reviewer's concern, we append the following words in the revised manuscript.

1. In the main text, we replace the numerical results of $m = 100$ with $m = 10$ in Fig 2 (b) of the original manuscript.
2. We have appended the numerical results with varying Schmidt rank and varying number of measurements to the Supplementary Material (i.e., SM E, Page 38).

Figure 5: Prediction error with various Schmidt rank and number of measurements.

References

- [1] Lov K Grover. A fast quantum mechanical algorithm for database search. In *Proceedings of the Twenty-eighth Annual ACM Symposium on Theory of Computing*, pages 212–219. ACM, 1996.
- [2] David Deutsch and Richard Jozsa. Rapid solution of problems by quantum computation. *Proceedings of the Royal Society of London. Series A: Mathematical and Physical Sciences*, 439(1907):553–558, 1992.
- [3] Aram W Harrow, Avinatan Hassidim, and Seth Lloyd. Quantum algorithm for linear systems of equations. *Physical review letters*, 103(15):150502, 2009.
- [4] Zheshen Zhang, Sara Mouradian, Franco NC Wong, and Jeffrey H Shapiro. Entanglement-enhanced sensing in a lossy and noisy environment. *Physical review letters*, 114(11):110506, 2015.
- [5] Michael R Grace, Christos N Gagatsos, and Saikat Guha. Entanglement-enhanced estimation of a parameter embedded in multiple phases. *Physical Review Research*, 3(3):033114, 2021.
- [6] Marco Piani and John Watrous. All entangled states are useful for channel discrimination. *Phys. Rev. Lett.*, 102:250501, Jun 2009.
- [7] Joonwoo Bae, Dariusz Chruściński, and Marco Piani. More entanglement implies higher performance in channel discrimination tasks. *Physical Review Letters*, 122(14):140404, 2019.
- [8] Hsin-Yuan Huang, Michael Broughton, Masoud Mohseni, Ryan Babbush, Sergio Boixo, Hartmut Neven, and Jarrod R McClean. Power of data in quantum machine learning. *Nature communications*, 12(1):1–9, 2021.
- [9] Yunchao Liu, Srinivasan Arunachalam, and Kristan Temme. A rigorous and robust quantum speed-up in supervised machine learning. *Nature Physics*, 17(9):1013–1017, 2021.
- [10] Jarrod R McClean, Sergio Boixo, Vadim N Smelyanskiy, Ryan Babbush, and Hartmut Neven. Barren plateaus in quantum neural network training landscapes. *Nature communications*, 9(1):1–6, 2018.
- [11] Marco Cerezo, Akira Sone, Tyler Volkoff, Lukasz Cincio, and Patrick J Coles. Cost function dependent barren plateaus in shallow parametrized quantum circuits. *Nature communications*, 12(1):1–12, 2021.
- [12] Michael Ragone, Bojko N Bakalov, Frédéric Sauvage, Alexander F Kemper, Carlos Ortiz Marrero, Martin Larocca, and M Cerezo. A unified theory of barren plateaus for deep parametrized quantum circuits. *arXiv preprint arXiv:2309.09342*, 2023.
- [13] Junyu Liu, Khadijeh Najafi, Kunal Sharma, Francesco Tacchino, Liang Jiang, and Antonio Mezzacapo. Analytic theory for the dynamics of wide quantum neural networks. *Physical Review Letters*, 130(15):150601, 2023.
- [14] Xuchen You and Xiaodi Wu. Exponentially many local minima in quantum neural networks. In *International Conference on Machine Learning*, pages 12144–12155. PMLR, 2021.

- [15] Xuchen You, Shouvanik Chakrabarti, and Xiaodi Wu. A convergence theory for over-parameterized variational quantum eigensolvers. *arXiv preprint arXiv:2205.12481*, 2022.
- [16] Xinbiao Wang, Junyu Liu, Tongliang Liu, Yong Luo, Yuxuan Du, and Dacheng Tao. Symmetric pruning in quantum neural networks. *arXiv preprint arXiv:2208.14057*, 2022.
- [17] Hsin-Yuan Huang, Richard Kueng, and John Preskill. Information-theoretic bounds on quantum advantage in machine learning. *Physical Review Letters*, 126(19):190505, 2021.
- [18] Marco Cerezo, Alexander Poremba, Lukasz Cincio, and Patrick J Coles. Variational quantum fidelity estimation. *Quantum*, 4:248, 2020.
- [19] Hsin-Yuan Huang, Richard Kueng, and John Preskill. Predicting many properties of a quantum system from very few measurements. *Nature Physics*, 16(10):1050–1057, 2020.
- [20] Andreas Elben, Steven T Flammia, Hsin-Yuan Huang, Richard Kueng, John Preskill, Benoît Vermersch, and Peter Zoller. The randomized measurement toolbox. *Nature Reviews Physics*, 5(1):9–24, 2023.
- [21] Angus Lowe and Ashwin Nayak. Lower bounds for learning quantum states with single-copy measurements. *arXiv preprint arXiv:2207.14438*, 2022.
- [22] Leonardo Banchi, Jason Pereira, and Stefano Pirandola. Generalization in quantum machine learning: A quantum information standpoint. *PRX Quantum*, 2(4):040321, 2021.
- [23] Leonardo Banchi, Jason Luke Pereira, Sharu Theresa Jose, and Osvaldo Simeone. Statistical complexity of quantum learning. *arXiv preprint arXiv:2309.11617*, 2023.
- [24] Matthias C Caro, Hsin-Yuan Huang, M Cerezo, Kunal Sharma, Andrew Sornborger, Lukasz Cincio, and Patrick J Coles. Generalization in quantum machine learning from few training data. *arXiv preprint arXiv:2111.05292*, 2021.
- [25] Hsin-Yuan Huang, Michael Broughton, Jordan Cotler, Sitan Chen, Jerry Li, Masoud Mohseni, Hartmut Neven, Ryan Babbush, Richard Kueng, John Preskill, et al. Quantum advantage in learning from experiments. *Science*, 376(6598):1182–1186, 2022.
- [26] Quntao Zhuang. Quantum ranging with gaussian entanglement. *Physical Review Letters*, 126(24):240501, 2021.
- [27] Matthias C Caro, Hsin-Yuan Huang, Nicholas Ezzell, Joe Gibbs, Andrew T Sornborger, Lukasz Cincio, Patrick J Coles, and Zoë Holmes. Out-of-distribution generalization for learning quantum dynamics. *arXiv preprint arXiv:2204.10268*, 2022.
- [28] Kunal Sharma, M Cerezo, Zoë Holmes, Lukasz Cincio, Andrew Sornborger, and Patrick J Coles. Reformulation of the no-free-lunch theorem for entangled datasets. *Physical Review Letters*, 128(7):070501, 2022.
- [29] Sitan Chen, Jordan Cotler, Hsin-Yuan Huang, and Jerry Li. Exponential separations between learning with and without quantum memory. In *2021 IEEE 62nd Annual Symposium on Foundations of Computer Science (FOCS)*, pages 574–585. IEEE, 2022.

- [30] Matthias C Caro, Elies Gil-Fuster, Johannes Jakob Meyer, Jens Eisert, and Ryan Sweke. Encoding-dependent generalization bounds for parametrized quantum circuits. *Quantum*, 5:582, 2021.
- [31] Xinbiao Wang, Yuxuan Du, Yong Luo, and Dacheng Tao. Towards understanding the power of quantum kernels in the NISQ era. *Quantum*, 5:531, 2021.
- [32] Alexander M Dalzell, Sam McArdle, Mario Berta, Przemyslaw Bienias, Chi-Fang Chen, András Gilyén, Connor T Hann, Michael J Kastoryano, Emil T Khabiboulline, Aleksander Kubica, et al. Quantum algorithms: A survey of applications and end-to-end complexities. *arXiv preprint arXiv:2310.03011*, 2023.
- [33] Yuxuan Du, Zhuozhuo Tu, Xiao Yuan, and Dacheng Tao. Efficient measure for the expressivity of variational quantum algorithms. *Physical Review Letters*, 128(8):080506, 2022.
- [34] Weikang Li and Dong-Ling Deng. Recent advances for quantum classifiers. *Science China Physics, Mechanics & Astronomy*, 65(2):220301, 2022.
- [35] Yuxuan Du, Yibo Yang, Dacheng Tao, and Min-Hsiu Hsieh. Problem-dependent power of quantum neural networks on multiclass classification. *Physical Review Letters*, 131(14):140601, 2023.
- [36] Yosep Kim, Yong-Su Kim, Sang-Yun Lee, Sang-Wook Han, Sung Moon, Yoon-Ho Kim, and Young-Wook Cho. Direct quantum process tomography via measuring sequential weak values of incompatible observables. *Nature communications*, 9(1):192, 2018.
- [37] Kyle Poland, Kerstin Beer, and Tobias J Osborne. No free lunch for quantum machine learning. *arXiv preprint arXiv:2003.14103*, 2020.

REVIEWERS' COMMENTS

Reviewer #2 (Remarks to the Author):

First, I would like to apologize for taking longer than usual to review the manuscript. I extend my thanks to all the authors for their careful consideration of my comments. In my opinion, the authors have significantly improved the quality of the manuscript by extending their results to the case of POVM and performing more numerical simulations. I am pleased to revise my initial assessment and now recommend publishing this manuscript.